Estimating vehicle carbon dioxide emissions from Boulder, Colorado using horizontal path-integrated column measurements

Eleanor M. Waxman[1], Kevin C. Cossel[1], Fabrizio Giorgetta[1], Gar-Wing Truong[1,2], William C. Swann[1], Ian Coddington[1], Nathan R. Newbury[1]

[1]Applied Physics Division, NIST Boulder
[2]Now at: Crystalline Mirror Solutions

Abstract

We performed seven and a half weeks of path-integrated concentration measurements of $CO_2$, $CH_4$, $H_2O$, and HDO over the city of Boulder, Colorado. An open-path dual-comb spectrometer simultaneously measured time-resolved data across a reference path, located near the mountains to the west of the city, and across an over-city path that intersected two-thirds of the city, including two major commuter arteries. By comparing the measured concentrations over the two paths when the wind is primarily out of the west, we observe daytime $CO_2$ enhancements over the city. Given the warm weather and the measurement footprint, the dominant contribution to the $CO_2$ enhancement is from city vehicle traffic. We use a Gaussian plume model combined with reported city traffic patterns to estimate city emissions of on-road $CO_2$ as $(6.2 \pm 2.2) \times 10^5$ metric tons (MT) $CO_2$/year, after correcting for non-traffic sources. Within the uncertainty, this value agrees with the city bottom-up greenhouse gas inventory for the on-road vehicle sector of $4.5 \times 10^5$ MT $CO_2$/year. Finally, we discuss experimental modifications that could lead to improved estimates from our path-integrated measurements.

1. Introduction

  Measurements of greenhouse gases, especially $CO_2$ and $CH_4$, are critical for monitoring, verification, and reporting as countries and cities work towards decreasing their carbon emissions. Measurements on the city-scale are critical because cities contribute to a large fraction of global emissions (Marcotullio et al., 2013; Seto et al., 2014). However, quantification of city greenhouse gas emissions is challenging, especially for $CO_2$ since it has a high background and numerous point and diffuse sources including traffic, power plants, and animal and plant respiration. Emissions of pollutants are typically determined using two methods: a top-down approach using atmospheric measurements over a specific site or area to adjust a prior model, and bottom-up inventories that calculate emissions based on sector activity and sector emissions factors. Here we demonstrate a technique for top-down measurements that uses an open-path sensor rather than a point sensor.

  Quantification of $CO_2$ fluxes from cities has been determined from eddy covariance flux measurements with a point sensor located on a tower in or near a city (Nemitz et al., 2002; Velasco et al., 2005; Coutts et al., 2007; Bergeron and Strachan, 2011; Velasco et al., 2014). However, for a single sensor, the relatively small footprint of the eddy covariance flux measurements limits the utility of this technique for large cities as do violations of the horizontal homogeneity assumption (Järvi et al., 2018). To overcome this limitation, tower networks of point sensors can measure $CO_2$ at multiple sites within a city and at background sites outside the city (McKain et al., 2012; Lauvaux et al., 2013; Bréon et al., 2015; Staufer et al., 2016; Lauvaux et al., 2016; Shusterman et al., 2016; Mueller et al., 2017; Verhulst et al., 2017; Sargent et al., 2018; Mitchell et al., 2018). To distinguish the small enhancements compared to the large background, these networks often use expensive, high-precision cavity ringdown (CRDS) instruments resulting in a high cost. The $BEACO_2N$ network (Shusterman et al., 2016), on the other hand, has a much lower cost per sensor. It requires calibration for quantitative results, but the high density of the point sensors can provide lower sensitivity to systematics (Turner et al., 2016). All of these methods use an inversion to determine the total emissions and thus rely on well-known priors and high-resolution mesoscale atmospheric models.

  More recently, several other approaches have also been applied to city-scale emissions. Aircraft mass balance measurements (White et al., 1976; Ryerson et al., 2001) have been used to determine city

emissions (Mays et al., 2009; Heimburger et al., 2017). However, the use of an aircraft is costly and labor
intensive, and therefore not suited to long-term continuous measurements. Column measurements from
the Total Carbon Column Observation Network (TCCON) were used to calculate total South Coast Air
Basin (SoCAB) CO and $CH_4$ emissions, but not $CO_2$ (Wunch et al., 2009). Data from the Orbiting
Carbon Observatory satellite (OCO-2) was recently combined with TCCON data to estimate $CO_2$
emissions from the LA basin (Hedelius et al., 2018).
As an alternative to these approaches, horizontal, kilometer-scale, open-path instruments could in
principle be used to determine $CO_2$ emissions from cities. Such instruments are capable of continuous
measurements over a large area with a single instrument, e.g.(Wong et al., 2016; Dobler et al., 2017;
Coburn et al., 2018). These sensors also have the advantage of being insensitive to small changes in local
meteorology and are not subject to the same representation errors as point sensors (Ciais et al., 2010).
Several such systems have been deployed. A laser absorption spectrometer system (GreenLITE) has
mapped $CO_2$ concentrations over Paris, but not yet quantified emissions (Dobler et al., 2017). The
California Laboratory of Atmospheric Remote Sensing Fourier Transform Spectrometer (CLARS-FTS) is
a downward-looking slant column Fourier transform spectrometer (FTS) that scans across 28
measurement targets in the Los Angeles Basin to measure $CO_2$, $CH_4$, and $O_2$ (Wong et al., 2015). Based
on the measured $CH_4$:$CO_2$ ratio and the bottom-up $CO_2$ inventory from California Air Resources Board,
researchers have calculated the LA Basin $CH_4$ emissions (Wong et al., 2016), but not yet the $CO_2$
emissions.
Here we present the quantification of city $CO_2$ emissions using open-path measurements made
with a dual frequency comb spectrometer. While dual-comb spectroscopy is a relatively new technique it
has a unique set of attributes that make it attractive for open path measurements (Rieker et al., 2014;
Coddington et al., 2016; Waxman et al., 2017; Coburn et al., 2018). Dual-comb spectroscopy (DCS) is a
high-resolution, broadband technique spanning hundreds of wavenumbers, but with a resolution that
exceeds even high-end FTIRs leading to a negligible instrument lineshape (Coddington et al., 2016). This
allows for simultaneous measurements of multiple species and path-integrated temperature with low
systematic uncertainty and without the need for instrument calibration. Additionally, the eye-safe, high-
brightness, single transverse-mode output of a frequency comb allows for beam paths exceeding 10 km
while the speed and parallelism of the measurement suppress any spectral distortion from the inevitable
turbulence-induced power fluctuations over such a path (Rieker et al., 2014; Waxman et al., 2017).
Figure 1 shows the measurement layout for an initial campaign to quantify $CO_2$ emissions from
Boulder, Colorado. Here we take the light from a dual comb spectrometer near the edge of the city and
simultaneously measure two paths: a reference path that points west-southwest towards the mountains and
an over-city path that crosses the city to the northeast, covering the main traffic arteries of the city with
sensitivity to traffic emissions. We acquire time-resolved data at 5-minute resolution of $CO_2$, $CH_4$, $H_2O$
and isotopologues over 7.5 weeks. The dry mole fraction of $CO_2$ shows a diurnal cycle consistent with a
morning build-up from traffic followed by a mid-day decline due to the rising boundary layer. In
addition, there is a distinct difference between the weekday and weekend cycles for $CO_2$, consistent with
traffic patterns. In order to demonstrate the utility of this method for emissions quantification, we perform
a preliminary estimate of the $CO_2$ emissions from traffic. To do this, we filter the data for days when the
wind is out of the west and not too strong so that there is a measurable daytime enhancement in $CO_2$
between the reference path and over-city path. Given the weather, beam path location, and observation
times, the dominant contribution will be from traffic rather than residential or industrial emissions. We
apply a Gaussian plume model to calculate the city emissions based on the expected distributed source
(due to traffic) and the path-averaged concentrations. After adjusting for small expected contributions
from residential sources and a local utility plant, the measured emission value is scaled to annual city-
wide emissions based on city traffic count data. We estimate $(6.2 \pm 2.2) \times 10^5$ metric tons (MT) $CO_2$/year,
compared to the bottom-up City of Boulder inventory estimate of $4.46 \times 10^5$ MT $CO_2$/year. Finally, we
discuss improvements to this estimate, which could be realized by more advantageous beam paths that
sample a larger spatial and temporal fraction of the full city emissions and by a more detailed inventory
model.
2. Experimental data
2.1 DCS measurements
The dual frequency comb spectroscopy (DCS) system was located on the top floor of the National
Institute of Standards and Technology (NIST) building in Boulder, Colorado. This instrument has been
described previously (Truong et al., 2016; Waxman et al., 2017). The light from the combs is split to
generate two combined dual-comb outputs, one of which is transmitted over the reference path and one of
which is transmitted over the city path (see Fig. 1.)  Here, we transmit 2-10 mW of light spanning 1.561
to 1.656 μm, which includes absorption lines from $CO_2$, $CH_4$, $H_2O$ and HDO. The returning light from
each path is detected and digitized to yield the transmitted optical spectrum at a point spacing of 0.0067
$cm^{-1}$ (1.5 picometer) and with effectively perfect (10 ppb) frequency accuracy and narrow instrument
lineshape (~$4 \times 10^{-6}$ $cm^{-1}$).  A typical spectrum from the reference path is shown in Fig. 2.  A fit of this
transmitted spectrum yields the path-averaged gas concentrations. The absolute frequency accuracy and
high frequency resolution of the dual-comb spectrometers translates to a high precision and accuracy in
the retrieved concentrations. Further, DCS spectra are undistorted by turbulence due to the simultaneous
acquisition of all spectral channels and the fast sample rate of the instrument (1.6 ms/spectrum, averaged
up to 5 minutes here) (Rieker et al., 2014).
In previous work (Waxman et al., 2017), we confirmed the high precision and accuracy possible
with open-path DCS. Two DCS instruments, constructed by different teams, measured atmospheric air
over adjacent paths over a two-week period. The retrieved path-averaged gas concentrations agreed to
better than 0.6 ppm (0.14%) for $CO_2$ and 7 ppb (0.35%) for $CH_4$ across the full two week period, where
the analysis of the two DCS instruments used a common spectral database (HITRAN 2008, Rothman et
al., 2009) to retrieve the concentrations from the absorption spectrum. In the work here, a single DCS
instrument probes the concentrations across two different open paths simultaneously, which should
further suppress any systematic offsets to below 0.45 ppm (Waxman et al., 2017). In addition, (Waxman
et al., 2017) compared the two DCS instruments to a stationary cavity ringdown (CRDS) point sensor
whose inlet was approximately at the midpoint of the open path.  This comparison actually took place
over the reference path during the first two weeks of the present work.  During that time, we found a
roughly constant difference of 3.4 ppm $CO_2$ and 17 ppb $CH_4$ between the DCS and CRDS systems.  At
present, we attribute this offset to differences in the calibration scheme as the DCS is tied to the HITRAN
database while the CRDS is tied to the manometric (or gravimetric depending on the gas) WMO scale.
Similar level offsets have been observed in comparison of the TCCON open-path FTS instrument and
point sensor-based vertical columns resulting in the TCCON $CO_2$ scaling factor of 0.9898 (4.08 ppm for a
mixing ratio of 400 ppm) (Wunch et al., 2015). This offset does not affect the results here as it is common
to both the reference and over-city paths.
The reference and over-city paths had different path lengths and therefore used slightly different
telescopes and launch powers. For the reference path, 2 mW of dual-comb light is launched from a 2-inch
home-built off-axis telescope (Cossel et al., 2017; Waxman et al., 2017). The light travels to a 2.5-inch
retroreflector located on a hilltop 1 km to the southwest of NIST and then is reflected back to a detector
that is co-located with the launch telescope for a 1950.17 ± 0.15 m round-trip path. Return powers vary
constantly with air turbulence but we collect about 200 μW for a typical 10 dB link loss.  For the city
path, 10 mW of dual-comb light is launched from a modified 10-inch diameter astronomical telescope to
a 5-inch retroreflector located on a building roof 3.35 km to the northeast for a 6730.66 ± 0.15 m round-
trip path.  We collect about 100 μW for a typical 20 dB link loss.  Round-trip path distances were
measured with a laser range finder.  Telescope tracking of the retroreflector is implemented to
compensate for thermal drifts via a co-aligned 850 nm light emitting diode (LED) and Silicon CCD
camera (Cossel et al., 2017; Waxman et al., 2017).
The measured spectra are analyzed as described in (Rieker et al., 2014; Waxman et al., 2017) at
32 second intervals.  Briefly, we fit a $7^{th}$-order polynomial and HITRAN data to the measured spectrum in
100-GHz (0.333 $cm^{-1}$) sections to remove the underlying structure from the comb themselves (as opposed

to the atmospheric absorption).  We fit the resulting absorption spectrum twice: once in the region from 6171 cm$^{-1}$ to 6271 cm$^{-1}$ (1.595 to 1.620 μm) to obtain the path-averaged temperature from the 1.6 μm $CO_2$ band, and once over the entire spectrum to obtain $^{12}CO_2$, $^{13}CO_2$, $CH_4$, $H_2O$, and HDO concentrations using the retrieved temperature. We then use the retrieved $H_2O$ concentration to correct the wet $CO_2$ and $CH_4$ mole fractions to dry mole fractions, hereafter referred to as $X_{CO2}$ and $X_{CH4}$ given in units of ppm and ppb (micromole of $CO_2$ per mole of dry air, and nanomole of $CH_4$ per mole of dry air).  The correction equations are $X_{CO2} = CO_2/(1-H_2O)$ and $X_{CH4} = CH_4/(1-H_2O)$.

The variations in the retrieved concentrations are due to statistical uncertainty, systematic uncertainty (discussed above), and the true variations in the gas concentrations.  Figure 8 of (Waxman et al., 2017) quantified the statistical uncertainty in terms of the Allan deviation over the 2-km reference path for both $X_{CH4}$ and $X_{CO2}$.  Figure 3 here provides an Allan deviation for just $X_{CO2}$ over both the ~6.7-km city and ~2-km reference paths, as calculated from a relatively "flat" 1000-s period of this measurement campaign on the night of 3 to 4 October 2016. As expected, the statistical uncertainty over both paths improves as the square root of integration time until reaching a floor, which we attribute to real variations in the atmospheric gas concentrations.  At 30 seconds, the statistical uncertainty of $X_{CO2}$is 0.76 ppm for the reference path and 0.64 ppm for the over-city path, finally dropping to 0.21 ppm and 0.15 ppm, respectively, at about 15 minutes.  In most subsequent figures, we show results at a 5-minute averaging time for which the statistical uncertainty is well under 0.3 ppm of $X_{CO2}$ for both paths and therefore well below the typical atmospheric variations. Note that the uncertainty also improves with path length, as expected due to the stronger absorption. The lower uncertainty over the city path reflects the expected improvement from the 3.4x longer path length lessened by the 2x reduction in return signal power for the longer path length.

## 2.2 Meteorological Measurements

Meteorological data including pressure, wind direction, and wind speed measurements are obtained from meteorological stations located at NCAR-Mesa and NCAR-Foothills (ftp://ftp.eol.ucar.edu/pub/archive/weather), which are approximately the endpoints of our measurement paths (see Fig. 1), as well as a 3-D sonic anemometer located at NIST.  The path-averaged air temperature was retrieved from the $CO_2$ spectra as described above.

## 2.3 Traffic data

We measure a subset of Boulder traffic, so we use the city traffic data to determine the fraction covered by our footprint (see Fig. 1).  Traffic data from the City of Boulder is freely available at: https://maps.bouldercolorado.gov/traffic-counts/?_ga=2.264109964.1414067815.1500302174-274759643.1492121882.  The city provides two types of traffic data that are useful in this work: the Arterial Count Program (ART) and the Turning Movement Count (TMC) data.

ART measures traffic at 18 major intersections in Boulder for five days (one work week, Monday through Friday) every year in one-hour bins to create a diurnal cycle. The traffic counts for 2016 are shown in Fig. 4. We use these data to scale our selected measurement time periods to a full day as discussed in section 3.3.4. Note that there is only a 10-20% "peak" in traffic counts at the standard commuter times with generally high traffic levels from 7:00 to ~19:00, which agrees with the traffic emissions reported by the Hestia inventory model for the similar city of Salt Lake City, UT (Mitchell et al., 2018).

TMC measures the number of vehicles at 140 intersections in Boulder for one work day per year during the hours of 7:45-8:45, 12:00-13:00, and 16:45-17:45. One third of each of these sites is measured every year. We have scaled the 2014 and 2015 data to 2016 traffic levels by using total vehicle mile values available from the City of Boulder. We approximate city vehicle emissions by using the TMC locations as our source locations with a source strength scaled based on the location's fractional traffic count.

## 3 Results and Discussion

3.1 DCS measurements
All 7.5 weeks of DCS measurements of $CO_2$, $CH_4$, $H_2O$, and HDO are shown in Fig. 5. HDO is
not used here but is shown for completeness (note that the HDO concentration is scaled by the isotopic
abundance in HITRAN). We have insufficient precision to measure time-resolved $^{13}CO_2$ concentrations
over the 2-km path. However, there are very clear enhancements in the over-city path relative to the
reference path for the other trace gases, especially for $CO_2$. These enhancements are observed primarily
at night when the boundary layer is lower. For example, on Oct. 13 the $CO_2$ enhancement reaches 129
ppm and the $CH_4$ enhancement reaches 265 ppb. Daytime enhancements occur when the wind speed is
very low and intermittent (typically below 5 m/s), which allows emitted gases to build up over the city.
When the wind increases to steady moderate speeds, the concentrations drop quickly as the emissions are
flushed out of the city. The $H_2O$ retrieval is important as accurate knowledge of the time-dependent water
concentration is needed to calculate the dry $CO_2$ and $CH_4$ mole fractions (see Section 2.1). Also, the
correlation of the water concentration between the two paths indicates the two paths sense the same air
mass, which is further substantiated in Figure 7a and is central to attributing their different CO2
concentration to local urban sources.
3.2 Diurnal Cycles
The diurnal cycle of $X_{CO2}$ and $X_{CH4}$ for both the reference path and the over city path are shown in
Fig. 6 for weekdays (midnight to midnight Monday through Friday) and weekends (midnight to midnight
Saturday and Sunday). We choose to include Monday as a weekday and Saturday as a weekend because
the influence of emissions from the previous day is expected to be low. The diurnal cycle of the wind
direction and the wind speed measured at NCAR Foothills are also shown in the top panel of Fig. 6. All
diurnal cycles are the median values over the full 7.5 weeks of measurements and the bars reflect the
25%/75% quartile values.
The diurnal cycle of the reference path $CO_2$ is nearly flat and nearly identical for both weekends
and weekdays. It has a slight maximum between 9 and 10 am, with average values of 410 to 420 ppm.
The diurnal cycle of the city path $CO_2$ shows a different trend with a stronger diurnal variation. Overnight
from about 6 pm (18:00) to 9 am, there is an enhancement in the $CO_2$ relative to the reference path as the
$CO_2$ from the city sources builds up due to the low winds out of the west and a presumed collapsing
nighttime boundary layer. During the weekdays, this enhancement increases in the morning consistent
with the rise in traffic. After the morning, the combination of the presumed rising boundary layer,
increased wind speed, and shift in average wind direction out of the west (270°) to the southeast (135 °)
result in a drop in the city path $CO_2$. Moreover, this shift in wind direction means that the reference path
no longer samples the clean air from the direction of the mountains but rather sees a very similar $CO_2$
enhancement as the city path. (Fortunately, as discussed below, there are days when the wind does not
shift direction so that there is a measured enhancement of the city path compared to the reference path.) In
the early evening, as the wind speed drops and the wind direction shifts back to out of the west, the
enhancement of the city path over the reference path reappears and continues overnight as the boundary
layer presumably drops. In general, the $CO_2$ mixing ratios tend to be higher on the weekdays, sometimes
exceeding 500 ppm, while weekend mixing ratios are entirely below 490 ppm. This difference is
reflected in the median values as well, which reach about 440 ppm during the weekdays but only 430 ppm
during the weekend.
The diurnal cycle of the reference path $CH_4$ is relatively flat for both weekends and weekdays at
just over 1.9 ppm, with a slight peak between 9 and 10 am. The diurnal cycle of the city path $CH_4$ shows
an enhancement, relative to the reference path, between midnight and about 9 am. We attribute this
enhancement to sources of $CH_4$ within the city combined again with low nighttime winds and collapsing
boundary layer. These sources may be leaking natural gas infrastructure such as observed in Boston
(Phillips et al., 2013; McKain et al., 2015; Hendrick et al., 2016), Washington, D.C. (Jackson et al.,
2014), and Indianapolis (Lamb et al., 2016). Unlike for $CO_2$, the $CH_4$ diurnal cycle appears unrelated to
traffic (nor would we expect it to be for clean-burning vehicles) as it does not increase during high traffic
times.
3.3 Estimate for $CO_2$ emissions due to traffic
3.3.1 Measurement day selections
To select test case days to estimate the city emissions, we filter the $X_{CO2}$ time series for time
periods with daytime enhancement and a moderate wind strength predominantly out of the west (270 °).
Given that the prevailing daytime winds are from the southeast (135°) and often strong, this limits the test
case days significantly. However, as is clear from Fig. 1, for these wind conditions, the city path samples
a significant fraction of the traffic emissions and the reference path samples no traffic emissions.  We
consider only daytime enhancements because the nighttime boundary layer behavior is significantly more
complicated than a well-mixed daytime stable boundary layer.  We find two days that meet these criteria:
Saturday 22 October 2016 from 11:00 to 16:00. and Tuesday 25 October 2016 from 7:00 to 16:00.  Both
days have moderate wind speeds (on average, 5 m/s) as measured at both meteorological sites.  There are
additional days with daytime enhancement in $X_{CO2}$, but the wind direction is variable. Additionally, there
are many days with no daytime enhancement in $X_{CO2}$ because the high wind speeds (6 m/s or higher)
prevented buildup of $CO_2$.  We use Oct. 22 as a proxy for all weekend days and Oct. 25 as a proxy for all
weekdays.  The $X_{CO2}$ and $X_{CH4}$ mixing ratios as well as wind speed and wind direction for these two case
study days are shown in Fig. 7.
In order to confirm that the reference path measured clean background air and the over-city path
measured city emissions, we calculated footprints for the two test case time periods using the Stochastic
Time-Inverted Lagrangian Transport (STILT-R) model (Fasoli et al., 2018). The input meteorology file
consisted of a uniform wind field with wind data from the NCAR Foothills lab, boundary layer height
from the North American Regional Reanalysis (NARR), uniform turbulent velocity variance calculated
from the Pasquill stability class (determined from wind speed and solar insolation) from the ground up to
the boundary layer, and the hyper near-field scaling described in Fasoli et al., (2018). Average footprints
for the two time periods are shown in Fig 7. The footprint for the reference path covers undeveloped areas
extending from the near foothills into the mountains. The footprint for the over-city path also has
contributions from the same general mountain region. In addition, this path has sensitivity to an extended
area within the city and therefore to a large fraction of the traffic emissions. Note the open-path geometry
leads to a much larger extended footprint for this path than would be the case for a single point sensor
located at the same height within the city.
The variability in the reference $CO_2$ on both days is a real atmospheric effect.  (In processing, any
data is removed if the signal power is low, which is indicative of poor telescope alignment or strong
weather-related attenuation over the beam path, so the variability is not due to variable signal strength.)
We attribute this variability to the smaller footprint of the reference path relative to the over-city path, as
seen in Fig. 7.  If the $CO_2$ in the air is not fully mixed, then the temporal and spatial variability will be
more evident in the path with the smaller footprint.
To convert from the measured enhancement to an emissions rate, we require a model that
connects the source strength to the plume concentration. Since we do not have a high-resolution, spatially
resolved inventory for Boulder similar to the Hestia model for Salt Lake City (Mitchell et al., 2018), we
use the existing Boulder traffic inventory (see Section 2.3) in conjunction with a Gaussian plume model.
3.3.2 Gaussian plume calculations
The standard Gaussian plume model that includes total reflection at the Earth's surface is
(Seinfeld and Pandis, 2006):
$$c(x, y, z, t) = \frac{q}{2\pi\sigma_y\sigma_z u} \exp\left(\frac{-(y-y_0)^2}{2\sigma_y^2}\right)\left[\exp\left(\frac{-(z-H)^2}{2\sigma_z^2}\right) + \exp\left(\frac{-(z+H)^2}{2\sigma_z^2}\right)\right] \qquad (1)$$
where $(x,y,z)$ is the location in space for which the plume concentration is being calculated, $(x_0, y_0, H)$ is the
emissions location, $c(x,y,z)$ is the concentration at location $(x,y,z)$ and time $t$, $q$ is the emissions strength
(usually in kg/s), $\sigma_y$ and $\sigma_z$ are the plume variances in the y and z direction as a function of travel distance
and Pasquill stability class (Seinfeld and Pandis, 2006), and $u$ is the wind speed in m/s. The wind is
assumed to be in the x-direction. The plume variances are calculated as:
$$\sigma_y = \exp\left[ I_y + J_y (\ln \Delta x) + K_y (\ln \Delta x)^2 \right] \qquad (2)$$
and
$$\sigma_z = \exp\left[ I_z + J_z (\ln \Delta x) + K_z (\ln \Delta x)^2 \right] \qquad (3)$$
where $I_y, J_y, K_y, I_z, J_z$, and $K_z$ are from a look-up table based on the Pasquill stability class, which depends
on the wind speed and solar insolation (Seinfeld and Pandis, 2006) and $\Delta x$ is the x-distance relative to the
plume origin. This plume model does not include any reflection at the boundary layer height; however,
due to the small spatial scales, this effect is negligible here.
We modify this equation in several ways: 1) Since we measure the column-integrated
concentration over a finite beam path at an angle to the wind direction, we integrate the plume
concentration along this beam path and then normalize to the length of the beam path. 2) We sum over
the emissions locations in the city that contribute emissions to our measurements. Thus our overall
measurement equation is:
$$(c - c_0) = \frac{Q}{L} \sum_{(x_j, y_j)} \int_0^L \frac{f_j}{2\pi \sigma_y \sigma_z u} \exp\left( \frac{-(s \sin\theta - y_j)^2}{2\sigma_y^2} \right) \left[ \exp\left( \frac{-(15-1)^2}{2\sigma_z^2} \right) + \exp\left( \frac{-(15+1)^2}{2\sigma_z^2} \right) \right] ds \quad (4)$$
where $(c - c_0)$ is our path-integrated concentration enhancement measurement (in MT/m$^3$ and MT is
metric tons; 1 MT = 1000 kg) along our path $s$ which goes from 0 to $L$, $Q$ is the total city emissions in
MT/hour, $L$ is our path length in m, $(x_j, y_j)$ are the source emissions locations, $f_j$ is the fraction of traffic at
source location $(x_j, y_j)$ relative to traffic over all locations in the city from the TMC database, $u$ is the wind
speed in m/s, $\theta$ is the angle of the beam path with respect to the wind direction, and $\sigma_y$ and $\sigma_z$ are the
plume dispersions in m in the y and z directions, which depend on the sources distance from the beam
path. In writing (4), we assume the wind is in the $+\hat{x}$ direction (which assumption is relaxed below). We
assume that all plume emission locations are vehicle tailpipes at 1 m above the ground, and the beam path
runs 15 m above ground so all measurement heights are at 15 m above ground.
*Grid rotation for variable wind directions*
To calculate (4), we grid the emissions locations using UTM (Universal Transverse Mercator)
coordinates obtained from Google Earth, where we then define north as $+\hat{y}$ and east as $+\hat{x}$. We translate
the coordinate system such that the DCS path begins at the origin (0,0) and travels a distance L at angle
$\theta$ with respect to the x-axis. Eq. (4) is then valid provided the wind is directly in the $+\hat{x}$ direction. More
generally, the wind is at a time varying small angle $\phi(t)$ with respect to $+\hat{x}$. Therefore, we apply a
rotation about the origin (Prussin et al., 2015):
$$\begin{bmatrix} \cos\phi & \sin\phi \\ -\sin\phi & \cos\phi \end{bmatrix} \begin{bmatrix} x \\ y \end{bmatrix} = \begin{bmatrix} x' \\ y' \end{bmatrix}$$
to generate new traffic coordinates $(x_j', y_j')$ and a new parameterized DCS beam path of $(s \cos(\theta'), s$
$\sin(\theta'))$ where $\theta' = \theta - \phi(t)$. In this new coordinate system, the wind is along the $+\hat{x}$ direction and Eq. (4)
holds with the substitutions $\theta \to \theta'$ and $y_j \to y_j'$, and where the $\sigma_y$ and $\sigma_z$ are calculated based on the
distance $\Delta x = |x_j' - (y_j'/\tan\theta')|$.
*Time dependent estimate of Q(t)*
The rotated Eq. *(4)* can be solved for $Q$ in terms of the measured or estimated values of $c(t)-c_0(t)$, $u(t)$,
$\Delta\phi(t)$, $\sigma_y(t)$, $\sigma_z(t)$, $\theta$, $L$, and $f_i$, where the first five quantities are time dependent. The resulting, time-
dependent $Q(t)$ for each test case day is shown in the bottom panels of Fig. 7 and has a mean value and
standard deviation of $Q_{Oct22} = 31 \pm 17$ MT $CO_2$/hour for October 22 and $Q_{Oct25} = 165 \pm 45$ MT $CO_2$/hour
for October 25 for the 5-minute averaged data as shown.
*Uncertainty in $Q(t)$*
Seven measured parameters factor in to the emissions calculation of $Q(t)$for the two days. These
are given in Table I along with the instrumental measurement precision and the observed variability. Note
that solar insolation is used solely in the determination of the Pasquill stability class (Seinfeld and
Pandis, 2006). The stability class is relatively insensitive to the variations in solar insolation observed on
the two test case days. As can be seen in the table, the uncertainty is dominated by the natural variability
in parameters like wind speed, wind direction, and $CO_2$ concentration rather than the DCS spectrometer
precision. The observed variability over the 5-9 hour period is typically at least a factor of 2 larger than
the instrument precision. The variability in these parameters leads to the observed variability in $Q(t)$. We
use the mean of $Q(t)$ as our emissions value and the standard deviation (at 5-minute time-averaging) as its
uncertainty. In using this standard deviation as a measure of the uncertainty, we attempt to capture the
uncertainty associated with the discrepancies between, for example, the weather-station measurements of
wind direction and speed relative to the true wind direction (which results in greater or fewer number of
plumes from the given traffic locations intercepting the measurement path). This variability appears in
$Q(t)$ as the nominal measured wind direction varies. Future systems with redundant, distributed DCS
beam paths would provide a superior estimate of all these uncertainties.
In addition, there are assumptions, and possible uncertainties, inherent to the Gaussian plume
model. First, the model does not include the effects of buildings, trees, or other objects that could break
up the plume between the emissions location and the beam path. Second, we assume that all $CO_2$
emissions come from the discrete locations shown in Fig. 1, while in reality the emissions are likely
substantially more diffuse. The assumption of discrete emissions simplifies modeling and is feasible due
to the city traffic data but may result in a bias due to the coarse distribution of traffic measurements.
Third, we approximate the measurement height at 15 m above ground although the beam height differs
over the path since Boulder is not perfectly flat. Finally, we use standard $I_y$, $J_y$, $K_y$, $I_z$, $J_z$, and $K_z$ values
which were derived for rural areas (Turner, 1970) which may be different than urban or suburban areas.
However, the greatest differences between rural and urban conditions are expected to be at night (Turner,
1970).
We further ran plume calculations in STILT-R using both wind fields derived from the local
meteorological stations shown in Figure 1 and using the North American Mesoscale Forecast System
(NAM, https://www.ncdc.noaa.gov/data-access/model-data/model-datasets/north-american-mesoscale-
forecast-system-nam). The High Resolution Rapid Refresh (HRRR, https://rapidrefresh.noaa.gov/hrrr/)
and North American Regional Reanalysis (NARR, https://www.ncdc.noaa.gov/data-access/model-
data/model-datasets/north-american-regional-reanalysis-narr) wind projections did not match the
measured winds at the meteorological stations. These calculations produced emissions values ranging
between 55 MT/hour and 770 MT/hour, depending on the wind fields and vertical dispersion
parameterization used. This brackets our emissions calculations by approximately a factor of three in
each direction and shows how sensitive these kilometer-scale measurements are to vertical dispersion.
3.3.3 Corrections for non-traffic sources of $CO_2$
There are a number of non-traffic sources of $CO_2$ that could contribute to our measured $X_{CO2}$
enhancement including local power plants, residential emission, and biological activity. These non-traffic
sources should have relatively minor contribution for several reasons. First, the footprint of the over-city
path does not overlap the large power plant to the east of the Boulder city limits. Second, the temperature
during the two test case days was 24 °C and 20 °C (68 °F and 75 °F) on October 22 and 25th leading to
minimal residential and commercial heating. Third, the measurements occurred in October after leaf
senescence so there should be negligible biological activity. Nevertheless, as discussed below, we do
adjust our measurements to account for the relatively minor contribution from non-traffic sources before
scaling up to an estimate of the annual traffic emissions.
We first consider power plants. There are two power generation facilities on the Department of
Commerce (DOC) campus located near the NIST building that houses the dual-comb spectrometer: the
site's Central Utilities Plant (CUP), and the National Oceanic and Atmospheric Administration (NOAA)
building's boilers.  To calculate their average $CO_2$ emissions, we used available fuel consumption data
(October 2016 monthly average for the CUP and mid-November to mid-December 2016 average for the
NOAA boilers; October data was unavailable) and the EPA emissions factor (EPA, 1995).  We then
modeled the CUP and boiler plume emissions using WindTrax (Flesch et al., 1995, 2004) with wind
speed and direction data from the NCAR-Mesa site.  We find that due to the moderate wind speeds (~5
m/s) during our case study days and the height mismatch between the emission stacks and our
measurement path over the DOC campus, there is negligible enhancement over the reference path.  Given
the location of the emission sources and the wind direction during our measurement periods, the
emissions also do not cross the over-city beam path.  Therefore, we apply no correction for these two
power plant emissions.
The University of Colorado also has a power plant that falls within the main footprint associated
with the over-city beam path, shown in Fig. 7a, and whose emissions are expected to intersect our over-
city beam path.  The EPA Greenhouse Gas Reporting Program (GHGRP,
https://www.epa.gov/ghgreporting) lists the 2017 emission from the power plant as $2.7 \times 10^4$ MT $CO_2$ or
an average of 3.1 MT/hour. (No breakdown by season or hour is provided.)  We apply this correction to
our previous daily values and add a conservative uncertainty equal to this correction in quadrature with
the previous uncertainty. The new adjusted values are then $28 \pm 17$ MT $CO_2$/hour for October 22 and 162
$\pm 45$ MT $CO_2$/hour for October 25.
The large Valmont power station lies just outside the city limits to the east of Boulder; however,
given its location and the dominant westerly wind, emissions from this source does not reach our beam
paths.  There are no other power generation facilities within the city that report to the GHGRP, so we
make no further corrections based on power plants.
In addition, there are also likely diffuse emissions from residential and commercial furnaces and
water heaters that use natural gas.  The City of Boulder Community Greenhouse Gas Emissions Inventory
reports twenty percent of the city emissions, or $3.18 \times 10^5$ MT CO2e, were from natural gas in 2016
(https://www-
static.bouldercolorado.gov/docs/2016_Greenhouse_Gas_Emissions_Inventory_Report_FINAL-1-
201803121328.pdf?_ga=2.130927943.970967930.1525795820-107394975).  The natural gas usage varies
strongly by month with building heating requirements. Although our measurements occurred in October,
the measurement days were quite warm (20-24 C) so that residential and commercial building heating
was unlikely and the use of an annual average would overestimate any contribution. Instead, we scale the
natural gas usage according to the monthly breakdown provided by the United States Energy Information
Administration database for Colorado (https://www.eia.gov/dnav/ng/hist/n3010co2m.htm). The mean
daytime (approximately sunrise to sunset, 7 am to 6 pm) temperature in October was 18.2 C while the
mean temperature (including day and night) for October was 15.7 C. Our daytime-only measurements
therefore had a mean temperature that was much closer to the mean temperature (day and night) of
September, which was 19.2 C. Therefore, we scale the Boulder annual natural gas consumption by the
September 2016 nature gas usage, which was 2.4% of the Colorado annual total according
(https://www.eia.gov/dnav/ng/hist/n3010co2m.htm). The estimated total emissions from residential and
commercial natural gas usage in Boulder over our measurement days is then 10.2 MT $CO_2$/hour. We
apply this correction to our measured values and include a (conservative) uncertainty equal to this
correction. The new adjusted values are then $Q_{Oct22,adj} = 18 \pm 20$ MT $CO_2$/hour for October 22 and $Q_{Oct25,adj}$
$= 152 \pm 46$ MT $CO_2$/hour for October 25.
Once leaf senescence has completed, neither plants nor soil respiration contribute to $CO_2$ signal
(Matyssek et al., 2013). The National Phenology Network (USA National Phenology Network, 2018)
data shows that for the site nearest to Boulder (64 km north of Boulder), the leaf fall dates were
September 15, 2016 for box elder trees October 6, 2016 for Eastern cottonwoods. Thus by our
measurement dates leaf senescence should be fully complete and plants will not contribute to the city $CO_2$
enhancement. We note that a wide range of biogenic contributions to $CO_2$ have been noted in the
literature (Gurney et al., 2017; Mitchell et al., 2018; Sargent et al., 2018).
3.3.4 Scaling to annual emissions
In order to compare with the city inventory, we scale our results to an annual total. To do this, we
use the hourly traffic data of Fig. 4 to scale $Q_{Oct22,adj}$ and $Q_{Oct25,adj}$ to a daily emission. Based on Figure 4,
34% of the total traffic counts occur during the 5-hour measurement period on Oct. 22 and 52% of the
total traffic counts occur during the 8-hour measurement period on Oct. 25 (excluding the 13:00 to 14:00
period). The daily emissions are then $Q_{Oct22,day} = Q_{Oct22,adj} \times (5 \text{ hours}) \div (0.34)$ and $Q_{Oct25,day} = Q_{Oct25,adj} \times (8$
hours)$\div(0.52)$ (The traffic data in Fig. 4 is based on weekday measurement and we assume that the
hourly distribution is the same for weekends; this may lead to a slight overestimate in the weekend data
where a larger fraction of emissions occurs between 11 am and 4 pm than on weekdays.) We then scale to
annual emissions by assuming that the emissions on Oct. 22 are representative of all 112 weekend/holiday
days and the emissions on Oct. 25 are representative of all 253 workdays. Including their uncertainty, this
calculation yields $(6.2 \pm 1.8) \times 10^5$ MT $CO_2$/year.
The scaling relies heavily on the traffic count data supplied by the city of Boulder, which does not
have an associated uncertainty value. A comparison of these data over several years shows a typical 7%
statistical variation at a given TMC location, after removing a linear trend. We assume this reflects day-
to-day fluctuations in traffic. In addition, there will be seasonal variations, which is not captured in the
extrapolation from our two test case days to the annual emissions. Due to the lack of seasonal data for
Boulder traffic, we use the detailed Hestia traffic inventory for Salt Lake City, UT given in Figure 2 of
(Mitchell et al., 2018). These data show a variation of ±18% in traffic emissions between "summer" and
"winter" months. Combined in quadrature with the 7% statistical uncertainty in the TMC traffic count
data, this leads to an additional ~20% uncertainty to the scaled annual estimate. As noted earlier, we have
not applied any additional uncertainty on the reliance on the TMC data as a proxy for emissions locations.
Including the additional uncertainty on the scaling to annual emissions, we estimate an annual
emission rate of $(6.2 \pm 2.2) \times 10^5$ MT $CO_2$/year for traffic carbon emissions for Boulder CO.
4 Comparison with city estimates
The city vehicle emissions estimate comes from total vehicle miles traveled based on data from
the transportation department, miles per gallon inputs from the EPA state inventory tool, and vehicle type
distribution from the Colorado Department of Public Health and the Environment (Kimberlee Rankin,
City of Boulder, personal communication).The City of Boulder estimates total vehicle emissions of
$4.50 \times 10^5$ metric tons (MT) of $CO_2$ in 2016 (https://www-
static.bouldercolorado.gov/docs/2016_Greenhouse_Gas_Emissions_Inventory_Report_FINAL-1-
201803121328.pdf?_ga=2.130927943.970967930.1525795820-107394975). On-road emissions account
for greater than 99% of the transportation emissions, so we have scaled this value down by one percent
for an on-road emissions value of $4.46 \times 10^5$ MT $CO_2$. We assume that all traffic emissions are $CO_2$ rather
than a mix of $CO_2$ and $CH_4$. There is no uncertainty provided by the city on this value.
In comparison, we estimate $(6.2 \pm 2.2) \times 10^5$ MT $CO_2$/year MT $CO_2$/year, which is 139% of the
city estimate but agrees within the given uncertainty. Interestingly, other studies have also found that
emissions measurements were higher than the reported inventory values. Brioude et al., (2013) found
top-down aircraft estimates of Los Angeles county and the South Coast Air Basin (SoCAB) $CO_2$ were
1.45 times larger than the Vulcan 2005 inventory (Gurney et al., 2009). An earlier aircraft campaign over
Sacramento, CA found an average $CO_2$ emission, with 100% uncertainty, that was 15-20% higher than
the Vulcan estimate (Turnbull et al., 2011). Lauvaux et al. (2016) compared Indianapolis city $CO_2$
emissions measured by a network of CRDS instruments to the HESTIA inventory (Gurney et al., 2012)
during INFLUX (Davis et al., 2017). They found that despite the building-scale resolution in the
HESTIA inventory, it still under-estimated the annual $CO_2$ flux by 20%. An updated version of HESTIA
predicted very similar emissions estimates for on-road, residential, and commercial sectors, so the
discrepancy was attributed to missing sources of $CO_2$, including animal (primarily human and companion
animal) respiration, biofuel combustion, and biosphere respiration (Gurney et al., 2017).
4.1 Improvements in future measurements
Future improvements should include additional and different beam paths, selected based on
prevailing wind directions. (Our initial assumption that the mountain path would generally act as a
reference path was incorrect since the prevailing daytime winds are not out of the west but rather the
southeast.) An east-west running beam north of the city and one south of the city would allow us to utilize
a larger fraction of the data as the predominant midday wind direction during the fall is out of the north to
north-east (see Fig. 1). Even longer beam paths would also interrogate a larger fraction of the city and
measure a correspondingly larger fraction of the vehicle emissions. Vertically-resolved data from e.g. a
series of stacked retroreflectors would better test the assumption of vertically-dispersing Gaussian
plumes.
Additionally, more extensive modeling to cover variable wind directions and speeds would allow
the incorporation of a much larger fraction of the data than the two days selected here. An inversion-
based model similar to (Lauvaux et al., 2013) could potentially be applied to a small city like Boulder;
however this would depend heavily on the quality of the bottom-up emissions inventory used to generate
the priors. Indeed, one of the major future improvements would be to generate a detailed Hestia inventory
of Boulder, CO similar to that generated for Salt Lake City, UT (Mitchell et al., 2018).
5 Conclusions
We demonstrate the use of an open-path dual frequency comb spectroscopy system for
quantifying city emissions of carbon dioxide. We send light over two paths: a reference path that
samples the concentration of gases entering the city from the west, and an over-city path that measures the
concentrations of gases after the air mass has crossed approximately two-thirds of the city including two
major commuter arteries. The measured diurnal cycle shows a significant traffic-related enhancement in
the carbon dioxide signal during weekdays in the over-city path compared to the reference path. We
select two case study days with appropriate wind conditions and apply Gaussian plume modeling to
estimate the total vehicular carbon emission. We then scale these results up to annual city-wide emissions
using traffic data from the City of Boulder. We find overall traffic related carbon emissions that are
approximately 1.4 times greater than the city's bottom-up traffic emissions inventory but with an
uncertainty that encompasses the city inventory estimate. Further improvements to this method should
include improved design of reference and over-city paths and a more detailed inventory model for
Boulder CO, which together should further reduce the overall uncertainty in the estimate.
Author contributions: EMW, KCC, IC, and NRN designed the experiment. WCS helped build the
hardware for the open-path measurements. EMW, KCC, and GWT ran the experiment. FG wrote the
processing code for the data analysis. EMW processed the data and did the Gaussian plume modeling.
KCC did the STILT-R modeling. EMW, KCC, IC, and NRN cowrote the manuscript.
Acknowledgements: We thank Kimberlee Rankin, Randall Rutsch, Bill Cowern, and Chris Hagelin from
the City of Boulder for city inventory and traffic information, Anna Karion for assistance with STILT-R
modeling, and Dave Plusquellic and Caroline Alden for assistance with the manuscript. This work was
funded by Defense Advanced Research Program Agency DSO SCOUT program, and James Whetstone
and the NIST special program office. Eleanor M. Waxman and Kevin C. Cossel are partially supported
by National Research Council postdoctoral fellowships.

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

Appendix A:  Modification of the Gaussian plume equation
Equation 1 is the standard Gaussian plume equation as discussed in Section 3.3.2 (Seinfeld and
Pandis, 2006).  It is reproduced here,
$$c(x,y,z,t) = \frac{q}{2\pi\sigma_y\sigma_z u}\exp\left(\frac{-(y-y_0)^2}{2\sigma_y^2}\right)\left[\exp\left(\frac{-(z-H)^2}{2\sigma_z^2}\right)+\exp\left(\frac{-(z+H)^2}{2\sigma_z^2}\right)\right]$$
where the standard variables are as defined in Section 3.3.2.
*Path-integrated substitutions*
The DCS returns the average concentration along a line path. We denote distance along this path
by the variable *s*, where *s* runs from 0 to *L*. This path is assumed to lie in the x-y plane at an
angle $\theta$ with respect to the x-axis (which is assumed to be the wind direction in the standard
Gaussian plume equation).  With these definitions, the contribution to the DCS signal from the
plume is,
$$(c-c_0) = \frac{1}{L}\int_0^L c(s\cos\theta,\ s\sin\theta,z,t)ds$$
or:

$$(c-c_0) = \frac{1}{L}\frac{q}{2\pi\sigma_y\sigma_z u}\int_0^L \exp\left(\frac{-(s\sin\theta-y_0)^2}{2\sigma_y^2}\right)\left[\exp\left(\frac{-(z-H)^2}{2\sigma_z^2}\right)+\exp\left(\frac{-(z+H)^2}{2\sigma_z^2}\right)\right]ds$$
*Accounting for multiple point sources*
Rather than a single source at $(x_0, y_0)$, we have multiple sources at locations $(x_j, y_j)$ , each with a
source strength $f_jq$, where $f_j$ is the fractional source strength out of the total value *q*. We now sum
over all sources to find the total enhancement. We also change the units of *q* from kg/s to
MT/year and thus change the emissions variable to *Q* to indicate the unit change.  This gives,
$$(c-c_0) = \frac{Q}{L}\sum_{(x_j,y_j)}\int_0^L \frac{f_j}{2\pi\sigma_y\sigma_z u}\exp\left(\frac{-(s\sin\theta-y_j)^2}{2\sigma_y^2}\right)\left[\exp\left(\frac{-(z-H)^2}{2\sigma_z^2}\right)+\exp\left(\frac{-(z+H)^2}{2\sigma_z^2}\right)\right]ds$$
*Height substitutions*
We assume that the point source emissions locations are 1 meter above ground ($z = 1$) and city
topographic data indicates that our beam path is approximately 15 meters above ground ($H = 15$).
These substitutions finally lead to Eq. (4) in the main text.

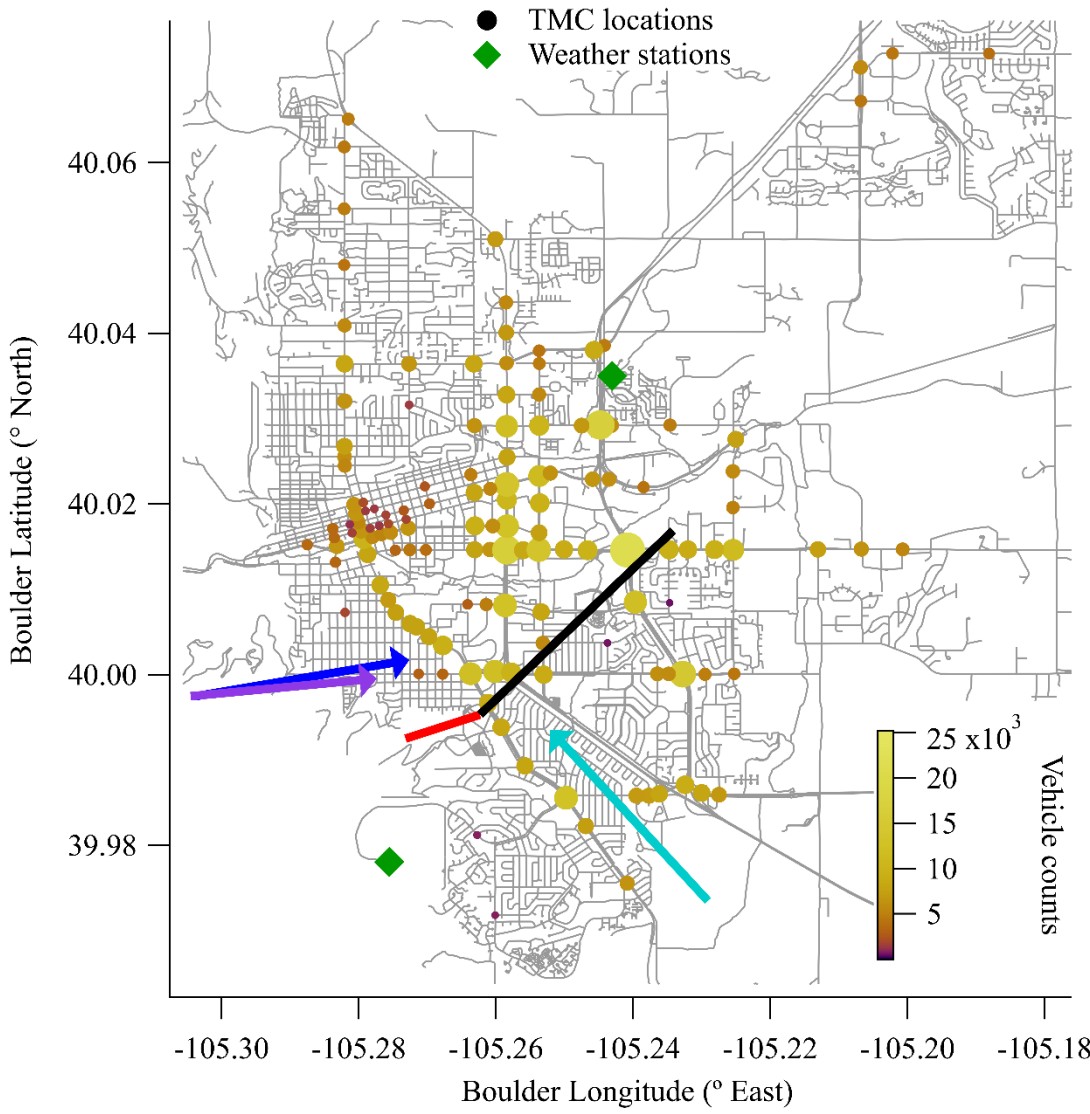

Figure 1: Measurement layout. The two measurement paths are shown by red (reference) and black
(over-city) lines. The two weather stations that provided wind speed and direction data are given by the
green diamonds. The colored circles are Turning Movement Count (TMC) locations, which are used as a
proxy for the traffic source locations. Both color and size represent the number of traffic counts at each
location. Dominant wind directions for the campaign overall (aqua) and the test case days (purple for
10/22 and blue for 10/25) are given by colored arrows.

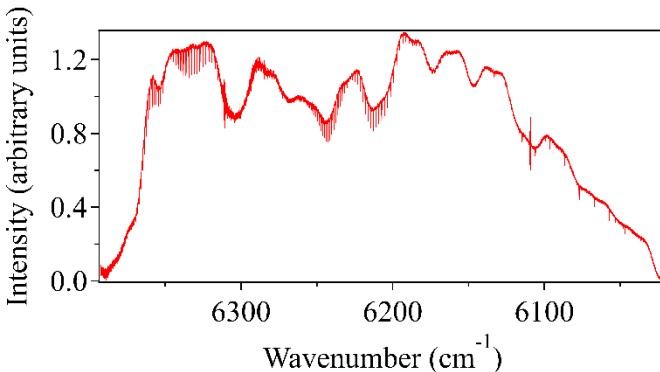

Figure 2:  Typical 32-second spectrum measured over the 2-km reference path.  $CO_2$ bands are observed
in the 6350 cm$^{-1}$ and 6225 cm$^{-1}$ regions, while $CH_4$ and $H_2O$ are measured between 6150 and 6050 cm$^{-1}$.
The larger, slowly varying structure is from the comb intensity profile. The atmospheric absorption
appears as the small and narrow dips.

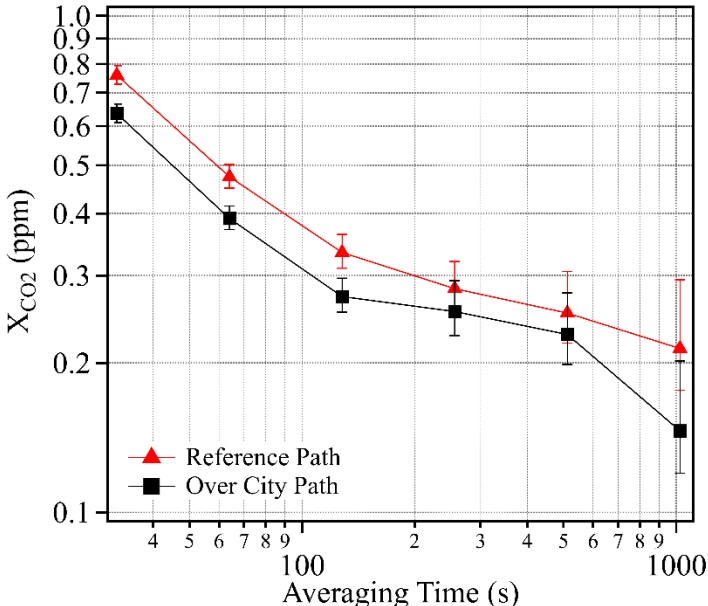

Figure 3:  Statistical uncertainty as quantified by the Allan deviations for $X_{CO2}$ over both the reference
path (red triangles) and city path (black squares) from a well-mixed, three-hour time period on the night
of October 3, 2016.

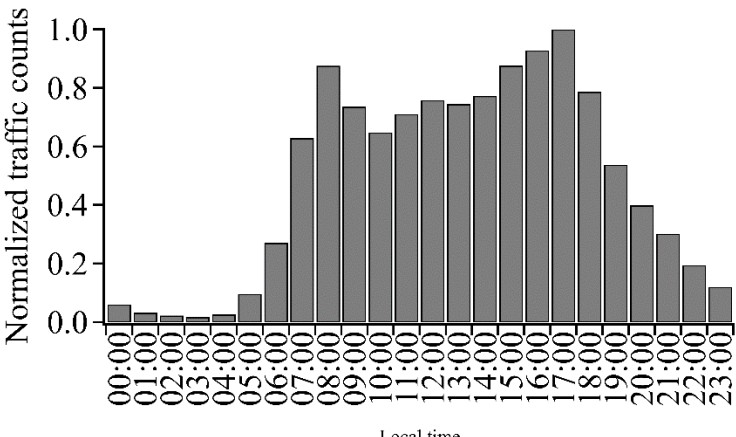

Figure 4: City-wide traffic counts from the Boulder Arterial Count Program (ART), normalized to a peak
of unity.

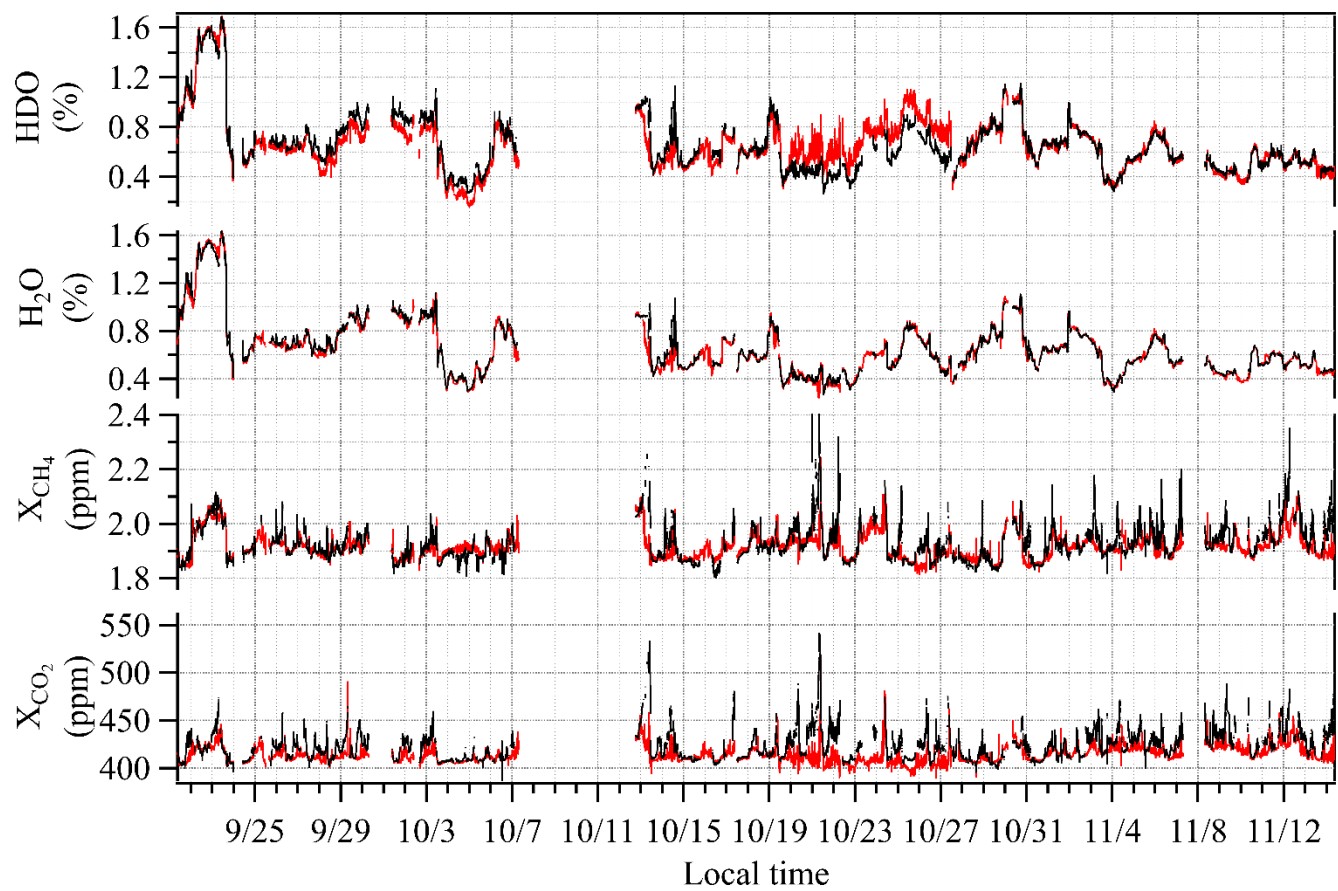

Figure 5: 7.5 weeks of dual-comb spectroscopy data for the reference path (red) and the over-city path
(black) smoothed to 5-minute time intervals. Enhancements in the over-city path relative to the reference
path are observed in $CO_2$ and $CH_4$ but not in $H_2O$ or HDO. (Note: the HDO concentration includes the
HITRAN isotopic scaling.)


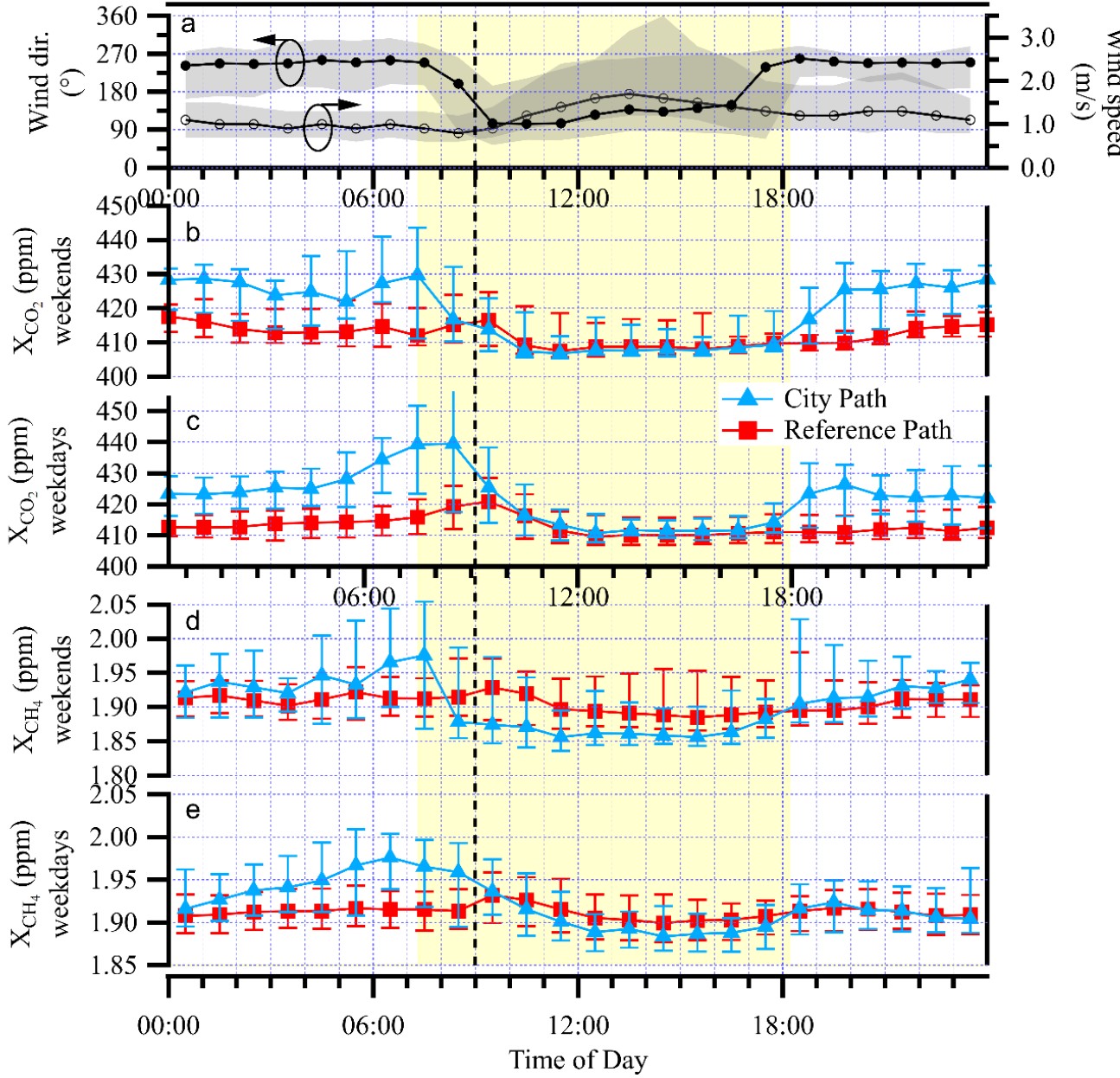

Figure 6: Diurnal cycle analysis. Data is the median of the full 7.5 weeks. (a) The mean direction in
which the wind is blowing (black trace, left axis) and wind speed (gray trace, right axis) both from the
NCAR Foothills measurement station, shaded regions reflect the 25th to 75th quartiles; (b) the weekend
and (c) weekday median $X_{CO_2}$ values for the over-city path (blue triangles) and reference path (red
squares). Uncertainty bars represent the 25%-75% range of values encountered. (d) and (e) Same data for
$X_{CH4}$. The vertical dashed black line marks 9:00 local time and the yellow shaded region highlights the
region from sunrise to sunset on Oct. 22, 2016.

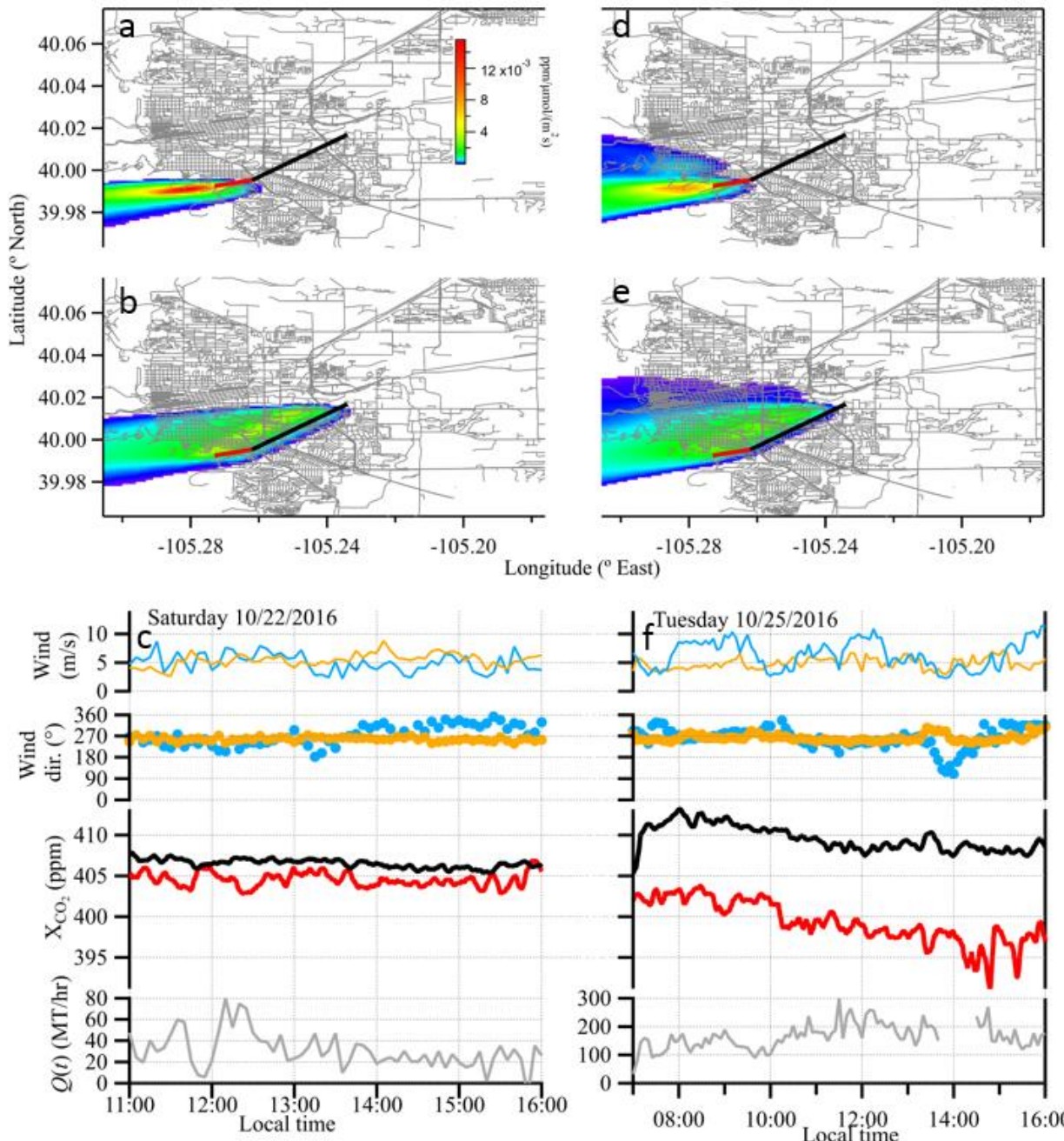

Figure 7: Footprint calculations and time series data for the two case study days. Left column: Saturday,
October 22, 2016; right column: Tuesday, October 25, 2016 data. Upper panels (a, d): Footprints for the
reference path. Middle panels (b, e): Footprints for the over-city path. The footprints are averaged over
the respective time windows and open paths. Lower panels (c,f): Wind and $CO_2$ data at 5-minute time
intervals. Reference and over-city measurement paths are shown in red and black, respectively. Data plots
show $X_{CO2}$ over the reference path (red) and city path (black), wind speed and wind direction measurements
taken at NCAR Mesa (blue) and NCAR Foothills (orange), and the calculated $Q(t)$. On Oct. 25, $Q(t)$ data
near 14:00 has been removed since the reference path wind direction is out of the southeast to east, resulting
in city contamination along the reference path. All data is smoothed to 5-minute time intervals.
Table I:  Parameters used to calculate the emission rate from Eq. (4).  The measurement precision refers
to the instrument uncertainty in the measurement quantity. The variability refers to the observed
environmental variability over the measurement period. The variability from the enhancement, the wind
direction, and the wind speed drive the observed variability in the estimated $Q(t)$ . (The distance from a
given source location to the DCS measurement path, $\Delta x_j$ , varies with location and has a 5-m uncertainty.)

| Quantity | Measurement precision | 10/22 11:00-16:00 | | 10/25 7:00-16:00 | |
|---|---|---|---|---|---|
| | | Mean | Variability | Mean | Variability |
| Pathlength $L$ | 0.15 m | 6730.66 m | 0 | 6730.66 m | 0 |
| Enhancement $(c\text{-}c_0)$ | 0.28 ppm (ref.) 0.25 ppm (city) | 1.99 ppm | 0.97 ppm (49%) | 10.3 ppm | 1.9 ppm (19%) |
| Wind speed $u$ | 0.3 m/s | 5.2 m/s | 1.0 m/s (19%) | 5.6 m/s | 1.3 m/s (23%) |
| Solar insolation | 5% | 570 W/m$^2$ | 76 W/m$^2$ (13%) | 275 W/m$^2$ | 185 W/m$^2$ (67%) |
| Wind direction $\phi$ | 2° | 265° | 21° | 264° | 15° |
