# Peer review of "Estimating vehicle carbon dioxide emissions from Boulder, Colorado using horizontal path-integrated column measurements"

_Atmospheric Chemistry and Physics, 2018_

## Referee Comment (RC1) · Anonymous Referee #1 · 5 Oct 2018

The open-path dual-comb spectrometer measurements are novel and could be of interest to numerous urban greenhouse gas (GHG) researchers. However, I see two major problems that preclude this paper from being publishable.

My major criticisms of the paper fall into two main categories: 1) Quality of measurements For a relatively new measurement technique, the paper is lacking in demonstration of measurement quality. Can these measurements be compared against nearby in-situ $CO_2$ observations? I believe the National Center for Atmospheric Research is carrying out $CO_2$ measurements on one of its buildings in Boulder (PI: Britton B. Stephens). NOAA-Earth System Research Laboratory (just next door to NIST, where

the authors are based) is the world leader for in-situ CO2 measurements, managing a wide network of CO2 measurement sites. Perhaps there are suitable measurement sites managed by NOAA-ESRL (e.g., a nearby tall tower) that can be used to compare against the horizontal column measurements shown in this paper?

The observational time series shown in Fig. 6 is quite noisy, it appears. The authors claim that the noise in the XCO2 reference path is associated with wind gusts of clean air, but no evidence is shown for this claim. Moreover, if the noise is due to gusts of clean air, why are there no such patterns showing up in the XCH4 time series? It is also concerning that the Q(t) time series does not appear to indicate a peak during the rush hour even on a weekday (10/25/2016).

Furthermore, the description of uncertainties associated with this technique is limited. There are scattered mentioning of the uncertainties—e.g., in Sect. 3.3.6. Instead, I believe a more substantial amount of text needs to be devoted to measurement uncertainties in the measurement section (Sect. 2.1). For instance, how does the uncertainty depend on path length and time resolution? Why?

2) Calculation of city-wide CO2 emissions The calculation of city-wide emissions is highly unsatisfactory. The authors attribute the CO2 signal as solely due to transportation, while multiple bottom-up inventories (e.g., Vulcan, Hestia) indicate that building emissions (mainly due to heating and cooking) are non-negligible. The building emissions are neglected entirely in this paper.

OTHER COMMENTS

What is the significance of the H2O and HDO measurements? They were shown, but no discussion is available for these species. What does HDO tell us? Are there any, say, meteorological events that can explain the variations in H2O and HDO?

Lines 34∼36: recent high profile papers published in PNAS for the cities of Boston (Sargent et al., 2018) and Salt Lake City (Mitchell et al., 2018) should also be mentioned

Lines 40∼41: another key limitation for urban eddy covariance measurements is the violation of horizontal homogeneity assumptions

Lines 69∼73: a reference for the CLARS instrument should be added

Line 127: typo; should be "turbulence"

Figure 1: The satellite map of Boulder is hard to decipher. Perhaps it would be better to use a road map instead, and zoomed in more? Also need to spell out "TMC" in the caption.

Figure 4: Variations in all 4 species are hard to discern. Are all 7.5 weeks needed? And could the y-axis range be reduced to show more variations? And is it necessary to show all 4 species in a single panel?

Figure 5: There are two different copies of this figure in the PDF file. Is the first one supposed to be deleted? Here the same problem exists—the variations in XCO2 and XCH4 are hard to figure out. I believe the variations in the median values are the most important for the reader. Currently the differences between the blue and red symbols indicating the median values are hardly visible. The y-axis ranges can be reduced considerably to highlight variations in the medians. Are the raw data really necessary here?

---

## Referee Comment (RC2) · Anonymous Referee #2 · 14 Oct 2018

Dear the authors of the manuscript:

The study made an attempt to estimate traffic CO2 emissions from the city of Boulder using CO2 data collected from a ground-based remote sensing instrument. The team has developed and maintained their open-path dual-comb spectrometer. The instrument has been subjected to several comparisons prior to this study. The authors conducted a 7.5-week long observation (September-November) and found two time periods (10/22 (Sat) 11am-4pm and 10/25 (Tue) 10am-4pm) they think suitable for the emission estimation of this study. The authors employed a Gaussian plume model with a city-wide traffic emission distribution constructed using the traffic data

collected in Boulder, in order to estimate the annual traffic emission from the city. The estimation yielded 6.9 x 10ˆ5 MT CO2/yr for the year 2016, which is 153% (155% in the manuscript) of the scaled city emission estimate (the 2015 traffic emission was scaled up using the total vehicle miles traveled in 2016). While several sources of errors and uncertainties due to the estimation approach were acknowledged, the authors discussed them by citing previous work, but they remained fully unquantified. The significance of this study is the application of CO2 data collected from the unique ground-based remote sensing instrument. But due to the poor design of the estimation approach and the lack of the evaluations of the results, I feel the authors failed to fully conclude this study and thus I do not recommend this manuscript for publication. I listed my major concerns and some other comments below.

1. Poor experimental design

I don't think the emission estimation was good enough to solely estimate traffic emissions. The authors had to make big assumptions to estimate annual traffic emissions from the city using two data period in 2016. Prior to the actual emission estimation, the authors needed to prove that they can get a reasonable estimation regardless of the assumptions made.

2. Traffic emissions

I checked the latest Boulder's inventory (2016) and their traffic emission was reduced by 10% from the previous year, which contradicts with the conclusion of this study and suggests a larger discrepancy between the authors' estimate and the inventory estimate.

City of Boulder Community Greenhouse Gas Emission Inventory https://www-static.bouldercolorado.gov/docs/2016_Greenhouse_Gas_Emissions_Inventory_Report_FINAL-1-201803121328.pdf?_ga=2.75749783.365316916.1539217252-1670325792.1539217252

[Figure]

Given the limited observation and the simple modeling, maybe it would have been a good idea to focus on the total city emission. I do not have an access to disaggregated CO2 emissions, but traffic emissions account for 28% of the city total emission, the residential is for 16% and commercial and industrial for 54%. Ignoring the contributions from other sectors does not seem to be a good idea, especially w/o doing any source attribution analysis.

3. Background problem?

This study used CO2 data from the reference path as a background. I understand that the air must be clean for the reference data, but my concern was the authors were comparing two different airmasses to calculate the CO2 enhancement. The only supporting information of background CO2 vs. city CO2 was the wind direction from a few observation points.

4. CO2-eq. I do not understand why the authors did not use emissions in the CO2 unit, rather than in the unit of CO2-eq. In collaboration with the city council, I would imagine it is not too difficult to obtain emission estimates solely for CO2.

5. Bottom-up vs. Top-down?

The discrepancy between bottom-up vs. top-down estimates are often large, as seen in previous studies. In many cases, the uncertainties associated with the inventory are assumed to be small. In this study, the authors have added a lot of potential errors when mapping the traffic emissions in space (approximated the spatial patterns) and time (scaled up to annual emission). Given that, less convincing than other studies if the authors did discussion just with citing papers.

Other comments:

P1, L31: "top-down measurement" sounds odd to me. How about "top-down approach using atmospheric measurements"?

P2, L58: A SoCAB CO2 top-down study has appeared on ACPD. Check out Hedelius

et al. (2018) ACPD.

P4, L162-: This traffic emission modeling is based on huge assumptions. The errors and uncertainties associated with this modeling needs to be quantified at least to show if the emission estimation approach in this study has a good accuracy to show the utility of the $CO_2$ data. Also, the authors might want to check the consistency/inconsistency between the traffic data the authors used and the new 2016 inventory used.

P6, L238: No correlations is good. . . but these are not shown anywhere.

P8, L321: 5% sounds small, but it is comparable to the potential emission changes we want to detect. I don't think it is insignificant.

P8, L327-: I think these are another set of big assumptions. These assumptions needed to be tested. The authors cited Gurney et al. (2017), but that is a case for Indianapolis. The authors could use the same logic for the large discrepancy between the top-down and bottom-up estimates. But the authors' statements are not supported by any quantitative analyses.

P16, Figure 1: This figure needs to be improved. It does not even clearly show the traffic distributions.

P19, Figure 4: Given the small changes in $CO_2$ we are discussing, I think the range of Y is too large. We can't see the variability in $CO_2$ data.

―――――――――――――――――――

---

## Author Comment (AC1) · 9 Feb 2019

*Dear Editor,*

*We thank the Reviewers for their thoughtful and thorough review of our manuscript. We appreciate the time and effort that they put into the reviews and we think that their comments have greatly improved our paper. We hope that we have sufficiently addressed the reviewer comments.*

*We have made substantial changes to the manuscript (please see detailed responses below), but the core of our paper has not changed and the emissions corrections suggested by the Reviewers have slightly improved the agreement with the City of Boulder results. We have modified the Introduction to incorporate the additional references suggested by the reviewers, as well as shorten it. We have added some additional details about the performance of the dual-comb spectrometer, based our previous work, and included a new figure of the Allan deviations. We have modified the Results and Discussion primarily to include additional uncertainty analysis and to incorporate a more thorough analysis of the non-$CO_2$ sources. We have updated Figure 1 and (now) 7 to incorporate reviewer comments including showing the footprints of the two paths based on a STILT-R calculation.*

*We have further made numerous small changes to the text and minor changes to all figures, such as making axes larger, to improve readability.*

*The reviewer comments are reproduced below in black. Our responses are below the reviewer comments in blue and italics. Our changes to the main text are in green. All line numbers refer to the clean version revised paper.*

*Sincerely,*

*Eleanor Waxman on behalf of the authors*

**Responses to specific reviewer comments:**

Reviewer 1:

The open-path dual-comb spectrometer measurements are novel and could be of interest to numerous urban greenhouse gas (GHG) researchers. However, I see two major problems that preclude this paper from being publishable.

My major criticisms of the paper fall into two main categories: 1) Quality of measurements For a relatively new measurement technique, the paper is lacking in demonstration of measurement quality. Can these measurements be compared against nearby in-situ $CO_2$ observations? I believe the National Center for Atmospheric Research is carrying out $CO_2$ measurements on one of its buildings in Boulder (PI: Britton B. Stephens). NOAA-Earth System Research Laboratory (just next door to NIST, where the authors are based) is the world leader for in-situ $CO_2$ measurements, managing a wide network of $CO_2$ measurement sites. Perhaps there are suitable measurement sites managed by NOAA-ESRL (e.g., a nearby tall tower) that can be used to compare against the horizontal column measurements shown in this paper?

*We have previously published work demonstrating the measurement quality of the DCS and apologize for not appropriately reviewing these results in this manuscript. In particular, we did the suggested comparison in a previously published work, Waxman et al. (2017) AMT. In addition to comparing two different DCS systems, we further compared them to a point sensor CRDS instrument (Picarro) measuring $CO_2$ and $CH_4$ whose inlet was located at approximately the midpoint of the path. This instrument was calibrated by NOAA GMD immediately after completing the measurement campaign discussed in this work. Note that in this comparison, we carefully located the point sensor along our "reference" measurement path. We think that this is an even better comparison than to the point sensor at NCAR, which is further away. Similarly, the nearest tall tower in operation during this campaign was the Boulder Atmospheric Observatory in Erie, Colorado which is approximately 22 km to the northeast of NIST in the middle of oil and gas fields. Due to the distance between the tall tower and our measurements, any comparison would have much larger uncertainty than the one already conducted in Waxman et al. (2017).*

*At lines 119-136, we have modified the text to say:*

In previous work (Waxman et al., 2017), we confirmed the high precision and accuracy possible with open-path DCS. Two DCS instruments, constructed by different teams, measured atmospheric air over adjacent paths over a two-week period. The retrieved path-averaged gas concentrations agreed to better than 0.6 ppm (0.14%) for $CO_2$ and 7 ppb (0.35%) for $CH_4$ across the full two week period, where the analysis of the two DCS instruments used a common spectral database (HITRAN 2008, Rothman et al., 2009) to retrieve the concentrations from the absorption spectrum. In the work here, a single DCS instrument probes the concentrations across two different open paths simultaneously, which should further suppress any systematic offsets to below 0.45 ppm (Waxman et al., 2017). In addition, (Waxman et al., 2017) compared the two DCS instruments to a stationary cavity ringdown (CRDS) point sensor whose inlet was approximately at the midpoint of the open path. This comparison actually took place over the reference path during the first two weeks of the present work. During that time, we found a roughly constant difference of 3.4 ppm $CO_2$ and 17 ppb $CH_4$ between the DCS and CRDS systems. At present, we attribute this offset to differences in the calibration scheme as the DCS is tied to the HITRAN database while the CRDS is tied to the manometric (or gravimetric depending on the gas) WMO scale. Similar level offsets have been observed in comparison of the TCCON open-path FTS instrument and point sensor-based vertical columns resulting in the TCCON $CO_2$ scaling factor of 0.9898 (4.08 ppm for a mixing ratio of 400 ppm) (Wunch et al., 2017). This offset does not affect the results here as it is common to both the reference and over-city paths.

The observational time series shown in Fig. 6 is quite noisy, it appears. The authors claim that the noise in the XCO2 reference path is associated with wind gusts of clean air, but no evidence is shown for this claim. Moreover, if the noise is due to gusts of clean air, why are there no such patterns showing up in the XCH4 time series? It is also concerning that the Q(t) time series does not appear to indicate a peak during the rush hour even on a weekday (10/25/2016).

*Figure 6 originally showed data at 30 second time resolution. There was no particular reason to show the data at this fine a time resolution and it adds white noise. We have updated the figure (now Figure 7) and all other figures to show data smoothed to 5 minute time resolution. In addition, we have further modified the axis range for the $CO_2$ on Oct. 22 to cover the same range as the Oct. 25 data and have removed the $CH_4$ time series as it is not relevant to the emissions analysis and Q(t) calculation.*

*We have looked more closely at the data in the old Figure 6 and agree with the reviewer that it is unlikely to be due to gusts of clean air. We now include representative footprints for the reference and over-city paths generated from STILT. We ascribe the greater variability along the reference path (red traces in Fig. 7) to the smaller footprint for this path as seen in Fig. 7. If the air entering the city is not fully mixed the spatial and temporal variability will be much more evident in the reference path because of the smaller footprint. Thus this is in a sense a representation error due to the path location selection. There are no major sources or sinks of $CH_4$ west of Boulder as opposed to $CO_2$ so we expect the $CH_4$ to be more well-mixed than the $CO_2$ and thus we do not expect such patterns in the $CH_4$ timeseries.*

*The modified Figure 7 is reproduced below.*

[Figure]

Figure 7: Footprint calculations and time series data for the two case study days. Left column: Saturday, October 22, 2016; right column: Tuesday, October 25, 2016 data. Upper panels (a, d): Footprints for the reference path. Middle panels (b, e): Footprints for the over-city path. The footprints are averaged over the respective time windows and open paths. Lower panels (c,f): Wind and $CO_2$ data at 5-minute time intervals. Reference and over-city measurement paths are shown in red and black, respectively. Data plots show $X_{CO2}$ over the reference path (red) and city path (black), wind speed and wind direction measurements taken at NCAR Mesa (blue) and NCAR Foothills (orange), and the calculated $Q(t)$. On Oct. 25, $Q(t)$ data near 14:00 has been removed since the reference path wind direction is out of the southeast to east, resulting in city contamination along the reference path. All data is smoothed to 5-minute time intervals.

*The modified the text at lines 289-291 read:*

We attribute this variability to the smaller footprint of the reference path relative to the over-city path, as seen in Fig. 7. If the $CO_2$ in the air is not fully mixed, then the temporal and spatial variability will be more evident in the path with the smaller footprint.

*The $Q(t)$ time series does not indicate a peak during rush hour because all of our data indicates that Boulder only has a very weak diurnal traffic cycle. It is a sufficiently small city that the authors can anecdotally say that it does not have rush hour the same way Los Angeles or Boston or New York City has a rush hour. The Boulder rush hour is a delay of a few minutes along a drive across town. As shown in the traffic count data of Figure 4, there is only a small (~10%) bump in the traffic counts, but generally the traffic is spread out fairly evenly from 8 am to 6 pm. Similarly, there is a similar weak (~10%) peak in the Salt Lake on road emissions according to Hestia (see Mitchell et al. 2018 PNAS, Figure 2) for the DBK site which has the population density closest to Boulder (1.5e3 people/$km^2$, 2010 census). Thus Boulder seems to be consistent with suburbs of other urban areas. We have modified the text at lines 191-194 to read:*

Note that there is only a 10-20% "peak" in traffic counts at the standard commuter times with generally high traffic levels from 7:00 to ~19:00, which agrees with the traffic emissions reported by the Hestia inventory model for the similar city of Salt Lake City, UT (Mitchell et al., 2018).

Furthermore, the description of uncertainties associated with this technique is limited. There are scattered mentioning of the uncertainties e.g., in Sect. 3.3.6. Instead, I believe a more substantial amount of text needs to be devoted to measurement uncertainties in the measurement section (Sect. 2.1). For instance, how does the uncertainty depend on path length and time resolution? Why?

*We have expanded the uncertainty discussion in Section 2.1 (and moved the previous text from Section 3.3.6 to this section). We have also included a new figure showing the Allan deviation for both the city and reference paths. We note that an analysis of the systematic uncertainty (e.g. effects of the optics, detection system, and fitting parameters) was also published in Waxman et al. (2017) AMT. The new paragraph in Section 2.1 starting at line 159 reads:*

The variations in the retrieved concentrations are due to statistical uncertainty, systematic uncertainty (discussed above), and the true variations in the gas concentrations. Figure 8 of (Waxman et al., 2017) quantified the statistical uncertainty in terms of the Allan deviation over the 2-km reference path for both $X_{CH4}$ and $X_{CO2}$. Figure 3 here provides an Allan deviation for just $X_{CO2}$ over both the ~6.7-km city and ~2-km reference paths, as calculated from a relatively "flat" 1000-s period of this measurement campaign on the night of 3 to 4 October 2016. As expected, the statistical uncertainty over both paths improves as the square root of integration time until reaching a floor, which we attribute to real variations in the atmospheric gas concentrations. At 30 seconds, the statistical uncertainty is 0.76 ppm for the reference path and 0.64 ppm for the over-city path, finally dropping to 0.21 ppm and 0.15 ppm, respectively, at about 15 minutes. In most subsequent figures, we show results at a 5-minute averaging time for which the statistical uncertainty is well under 0.3 ppm of $X_{CO2}$ for both paths and therefore well below the typical atmospheric variations. Note that the uncertainty also improves with path length, as expected due to the strong absorption. The lower uncertainty over the city path reflects the expected improvement from the 3.4x longer path length lessened by the 2x reduction in return signal power also due to the longer path length.

*We have also included a new Figure 3 which is the Allan deviation for both the reference and over-city paths during a three-hour time period on the night of October 3-4 which is rather flat for both paths:*

[Figure]

Figure 3: Statistical uncertainty as quantified by the Allan deviations for $X_{CO2}$ over both the reference path (red triangles) and city path (black squares) from a well-mixed, three-hour time period on the night of October 3, 2016.

2) Calculation of city-wide CO2 emissions The calculation of city-wide emissions is highly unsatisfactory. The authors attribute the CO2 signal as solely due to transportation, while multiple bottom-up inventories (e.g., Vulcan, Hestia) indicate that building emissions (mainly due to heating and cooking) are non-negligible. The building emissions are neglected entirely in this paper.

*In response to this comment and the one from Reviewer 2, Comment 2 we have rewritten Section 3.3.3 to include a thorough analysis of non-traffic sources of $CO_2$ during our measurement time period. Unfortunately, Hestia is not yet available for Boulder and Vulcan 2.2 is only available for 2002 (14 years prior to the present study). Therefore, we cannot extract Vulcan vehicle emissions estimates for Boulder. However, we address this using the City of Boulder greenhouse gas inventory data from 2016 (*[https://www-static.bouldercolorado.gov/docs/2016_Greenhouse_Gas_Emissions_Inventory_Report_FINAL-1-201803121328.pdf?_ga=2.130927943.970967930.1525795820-107394975](https://www-static.bouldercolorado.gov/docs/2016_Greenhouse_Gas_Emissions_Inventory_Report_FINAL-1-201803121328.pdf?_ga=2.130927943.970967930.1525795820-107394975)*). This results in a correction of 14% of our calculated emission value. We have added text at lines 412-433 in Section 3.3.3 on residential emissions:*

In addition, there are also likely diffuse emissions from residential and commercial furnaces and water heaters that use natural gas. The City of Boulder Community Greenhouse Gas Emissions Inventory reports twenty percent of the city emissions, or $3.18\times10^5$ MT CO2e, were from natural gas in 2016 (https://www-static.bouldercolorado.gov/docs/2016_Greenhouse_Gas_Emissions_Inventory_Report_FINAL-1-201803121328.pdf?_ga=2.130927943.970967930.1525795820-107394975). The natural gas usage varies strongly by month with building heating requirements. Although our measurements occurred in October, the measurement days were quite warm (20-24 C) so that residential and commercial building heating was unlikely and the use of an annual average would overestimate any contribution. Instead, we scale the

natural gas usage according to the monthly breakdown provided by the United States Energy Information Administration database for Colorado (https://www.eia.gov/dnav/ng/hist/n3010co2m.htm). The mean daytime (approximately sunrise to sunset, 7 am to 6 pm) temperature in October was 18.2 C while the mean temperature (including day and night) for October was 15.7 C. Our daytime-only measurements therefore had a mean temperature that was much closer to the mean temperature (day and night) of September, which was 19.2 C. Therefore, we scale the Boulder annual natural gas consumption by the September 2016 nature gas usage, which was 2.4% of the Colorado annual total according (https://www.eia.gov/dnav/ng/hist/n3010co2m.htm). The estimated total emissions from residential and commercial natural gas usage in Boulder over our measurement days is then 10.2 MT $CO_2$/hour. We apply this correction to our measured values and include a (conservative) uncertainty equal to this correction. The new adjusted values are then $Q_{Oct22,adj} = 18 \pm 20$ MT $CO_2$/hour for October 22 and $Q_{Oct25,adj} = 152 \pm 46$ MT $CO_2$/hour for October 25.

**OTHER COMMENTS**

What is the significance of the H2O and HDO measurements? They were shown, but no discussion is available for these species. What does HDO tell us? Are there any, say, meteorological events that can explain the variations in H2O and HDO?

*The H2O measurements are required here to extract the dry mole fraction. The main significance for this work is that both paths see the same variations in these quantities, further supporting the claim that the same general air mass is sampled by the two paths so that the enhancement of carbon dioxide is attributed to local sources rather than remote sources. The variations are attributed to the normal humidity variations expected from general weather patterns but we have not conducted an analysis of these. Finally, we include HDO since it indicates the multispecies sensing capabilities of the DCS and might be useful in future analysis, but it is true that the HDO variations are not discussed at length here and the additional data plot is not strictly necessary. Discussion of the H₂O/HDO ratios and an accompanying meteorology analysis is outside the scope of this work.*

*We have added the following text at lines 205-207:* HDO is not used here but is shown for completeness (note that the HDO concentration is scaled by the isotopic abundance in HITRAN).

*And lines 214-215:* The $H_2O$ retrieval is important as accurate knowledge of the time-dependent water concentration is needed to calculate the dry $CO_2$ and $CH_4$ mole fractions (see Section 2.1).

Lines 34~36: recent high profile papers published in PNAS for the cities of Boston (Sargent et al., 2018) and Salt Lake City (Mitchell et al., 2018) should also be mentioned

*We have included these references at line 43.*

Lines 40~41: another key limitation for urban eddy covariance measurements is the violation of horizontal homogeneity assumptions

*We thank the reviewer for the comment and have added it to the text at lines 38-39 that now reads:* "…utility of this technique for large cities as do violations of the horizontal homogeneity assumption (Järvi et al., 2018)."

Lines 69~73: a reference for the CLARS instrument should be added

*We have cited Wong et al. (2015) ACP, "Mapping CH4:$CO_2$ ratios in Los Angeles with CLARS-FTS from Mount Wilson, California" at lines 66.*

Line 127: typo; should be "turbulence"

*We have corrected this.*

Figure 1: The satellite map of Boulder is hard to decipher. Perhaps it would be better to use a road map instead, and zoomed in more? Also need to spell out "TMC" in the caption.

*We have modified the figure as suggested. In addition and in response to a similar comment by the Reviewer 2, we have changed the size of the TMC traffic symbols to indicate the traffic count. The updated figure and caption are as follows:*

[Figure]

Figure 1: Measurement layout. The two measurement paths are shown by red (reference) and black (over-city) lines. The two weather stations that provided wind speed and direction data are given by the green diamonds. The colored circles are Turning Movement Count (TMC) locations, which are used as a proxy for the traffic source locations. Both color and size represent the number of traffic counts at each location. Dominant wind directions for the campaign overall (aqua) and the test case days (purple for 10/22 and blue for 10/25) are given by colored arrows.

Figure 4: Variations in all 4 species are hard to discern. Are all 7.5 weeks needed? And could the y-axis range be reduced to show more variations? And is it necessary to show all 4 species in a single panel?

*We would prefer to retain all 7.5 weeks since it establishes that the DCS system is capable of this level of continuous operation, which will be needed if DCS is ever to be incorporated into long-term urban*

*monitoring. The y-axis has been expanded almost to the peak-to-peak variations already, unfortunately. To address the reviewers valid comment, though, we have changed the aspect ratio of the figure by approximately a factor of 2. We have also smoothed the data to 5 minute integration time in order to suppress the statistical uncertainty (see Section 2.1) and enhance the true variations and differences measured across the two paths:*

[Figure]

Figure 5: 7.5 weeks of dual-comb spectroscopy data for the reference path (red) and the over-city path (black) smoothed to 5-minute time intervals. Enhancements in the over-city path relative to the reference path are observed in $CO_2$ and $CH_4$ but not in $H_2O$ or HDO. (Note: the HDO concentration includes the HITRAN isotopic scaling.)

Figure 5: There are two different copies of this figure in the PDF file. Is the first one supposed to be deleted? Here the same problem existsâ˘Tthe variations in XCO2 and ˘ XCH4 are hard to figure out. I believe the variations in the median values are the most important for the reader. Currently the differences between the blue and red symbols indicating the median values are hardly visible. The y-axis ranges can be reduced considerably to highlight variations in the medians. Are the raw data really necessary here?

*As suggested by the reviewer, we have removed the raw data. However, the raw data has such a large range that we still indicate this via the uncertainty bars on the median points, which reflect the 25%-75% range of the values encountered. The y-axis has been expanded to the largest possible range that still captures this uncertainty. Following the comments of the reviewer, the revised figure is inserted below:*

[Figure]

Figure 6: Diurnal cycle analysis. Data is the median of the full 7.5 weeks. (a) The mean direction in which the wind is blowing (black trace, left axis) and wind speed (gray trace, right axis) both from the NCAR Foothills measurement station, shaded regions reflect the 25th to 75th quartiles; (b) the weekend and (c) weekday median $X_{CO2}$ values for the over-city path (blue triangles) and reference path (red squares). Uncertainty bars represent the 25%-75% range of values encountered. (d) and (e) Same data for $X_{CH4}$. The vertical dashed black line marks 9:00 local time and the yellow shaded region highlights the region from sunrise to sunset on Oct. 22, 2016.

Reviewer 2:

Dear the authors of the manuscript:

The study made an attempt to estimate traffic CO2 emissions from the city of Boulder using CO2 data collected from a ground-based remote sensing instrument. The team has developed and maintained their open-path dual-comb spectrometer. The instrument has been subjected to several comparisons prior to this study. The authors conducted a 7.5-week long observation (September-November) and found two time periods (10/22 (Sat) 11am-4pm and 10/25 (Tue) 10am-4pm) they think suitable for the emission estimation of this study. The authors employed a Gaussian plume model with a city-wide traffic emission distribution constructed using the traffic data collected in Boulder, in order to estimate the annual traffic emission from the city. The estimation yielded 6.9 x 10^5 MT CO2/yr for the year 2016, which is 153% (155% in the manuscript) of the scaled city emission estimate (the 2015 traffic emission was scaled up

using the total vehicle miles traveled in 2016). While several sources of errors and uncertainties due to the estimation approach were acknowledged, the authors discussed them by citing previous work, but they remained fully unquantified. The significance of this study is the application of CO2 data collected from the unique ground-based remote sensing instrument. But due to the poor design of the estimation approach and the lack of the evaluations of the results, I feel the authors failed to fully conclude this study and thus I do not recommend this manuscript for publication. I listed my major concerns and some other comments below.

1.Poor experimental design

I don't think the emission estimation was good enough to solely estimate traffic emissions. The authors had to make big assumptions to estimate annual traffic emissions from the city using two data period in 2016. Prior to the actual emission estimation, the authors needed to prove that they can get a reasonable estimation regardless of the assumptions made.

*We agree with the Reviewer that we had to make large assumptions. In the revised manuscript, we have tried to be clear about those assumptions and include some estimated uncertainties. Specifically, we hope we have addressed the Reviewer's comments by 1) discussing the uncertainty of the DCS in the revised Section 2.1, 2) adding a new Section 3.3.3 that discusses the non-traffic sources and uncertainties, and 3) combining and elaborating on the scaling uncertainties in Section 3.3.4. Scaling to annual emissions is necessary as it is the only way to compare to the bottom-up inventory for the city, as we do not have an up-to-date time-resolved inventory like Vulcan or Hestia, as discussed in response to Reviewer 1 above in comment 2. Section 3.3.4 is reproduced below:*

3.3.4 Scaling to annual emissions

In order to compare with the city inventory, we scale our results to an annual total. To do this, we use the hourly traffic data of Fig. 4 to scale $Q_{Oct22,adj}$ and $Q_{Oct25,adj}$ to a daily emission. Based on Figure 4, 34% of the total traffic counts occur during the 5-hour measurement period on Oct. 22 and 52% of the total traffic counts occur during the 8-hour measurement period on Oct. 25 (excluding the 13:00 to 14:00 period). The daily emissions are then $Q_{Oct22,day} = Q_{Oct22,adj} \times (5 \text{ hours}) \div (0.34)$ and $Q_{Oct25,day} = Q_{Oct25,adj} \times (8 \text{ hours}) \div (0.52)$ (The traffic data in Fig. 4 is based on weekday measurement and we assume that the hourly distribution is the same for weekends; this may lead to a slight overestimate in the weekend data where a larger fraction of emissions occurs between 11 am and 4 pm than on weekdays.) We then scale to annual emissions by assuming that the emissions on Oct. 22 are representative of all 112 weekend/holiday days and the emissions on Oct. 25 are representative of all 253 workdays. Including their uncertainty, this calculation yields $(6.2 \pm 1.8) \times 10^5$ MT $CO_2$/year.

The scaling relies heavily on the traffic count data supplied by the city of Boulder, which does not have an associated uncertainty value. A comparison of these data over several years shows a typical 7% statistical variation at a given TMC location, after removing a linear trend. We assume this reflects day-to-day fluctuations in traffic. In addition, there will be seasonal variations, which is not captured in the extrapolation from our two test case days to the annual emissions. Due to the lack of seasonal data for Boulder traffic, we use the detailed Hestia traffic inventory for Salt Lake City, UT given in Figure 2 of (Mitchell et al., 2018). These data show a variation of ±18% in traffic emissions between "summer" and "winter" months. Combined in quadrature with the 7% statistical uncertainty in the TMC traffic count data, this leads to an additional ~20% uncertainty to the scaled annual estimate. As noted earlier, we have not applied any additional uncertainty on the reliance on the TMC data as a proxy for emissions locations. Including the additional uncertainty on the scaling to annual emissions, we estimate an annual emission rate of $(6.2 \pm 2.2) \times 10^5$ MT $CO_2$/year for traffic carbon emissions for Boulder CO.

*We agree that longer time series data and different path locations would reduce the required assumptions. We discuss this in Section 4.1 in the revised manuscript:*

4.1 Improvements in future measurements

Future improvements should include additional and different beam paths, selected based on prevailing wind directions. (Our initial assumption that the mountain path would generally act as a reference path was incorrect since the prevailing daytime winds are not out of the west but rather the southeast.) An east-west running beam north of the city and one south of the city would allow us to utilize a larger fraction of the data as the predominant midday wind direction during the fall is out of the north to north-east (see Fig. 1). Even longer beam paths would also interrogate a larger fraction of the city and measure a correspondingly larger fraction of the vehicle emissions. Vertically-resolved data from e.g. a series of stacked retroreflectors would better test the assumption of vertically-dispersing Gaussian plumes.

Additionally, more extensive modeling to cover variable wind directions and speeds would allow the incorporation of a much larger fraction of the data than the two days selected here. An inversion-based model similar to (Lauvaux et al., 2013) could potentially be applied to a small city like Boulder; however this would depend heavily on the quality of the bottom-up emissions inventory used to generate the priors. Indeed, one of the major future improvements would be to generate a detailed Hestia inventory of Boulder, CO similar to that generated for Salt Lake City, UT (Mitchell et al., 2018).

2. Traffic emissions

I checked the latest Boulder's inventory (2016) and their traffic emission was reduced by 10% from the previous year, which contradicts with the conclusion of this study and suggests a larger discrepancy between the authors' estimate and the inventory estimate.

*We have updated the paper to compare directly to the 2016 inventory of $4.46 \times 10^5$ MT/year (as opposed to the value used in the original submitted version of $4.52 \times 10^5$ MT/year). As a note, the 10% reduction the reviewer cites is likely the reduction from 2005 to 2016, not 2015 to 2016. The figure from the city inventory comparison is reproduced below:*

[Figure]

*From https://www-static.bouldercolorado.gov/docs/2016_Greenhouse_Gas_Emissions_Inventory_Report_FINAL-1-201803121328.pdf?_ga=2.130927943.970967930.1525795820-1073949754.1525187370*

*We have edited the text at lines 470-475 to read:*

The City of Boulder estimates total vehicle emissions of $4.50 \times 10^5$ metric tons (MT) of $CO_2$ in 2016 (https://wwwstatic.bouldercolorado.gov/docs/2016_Greenhouse_Gas_Emissions_Inventory_Report_FINAL-1-201803121328.pdf?_ga=2.130927943.970967930.1525795820-107394975). On-road emissions account for greater than 99% of the transportation emissions, so we have scaled this value down by one percent for an on-road emissions value of $4.46 \times 10^5$ MT $CO_2$.

City of Boulder Community Greenhouse Gas Emission Inventory
https://wwwstatic.bouldercolorado.gov/docs/2016_Greenhouse_Gas_Emissions_Inventory_Report_FINAL1-201803121328.pdf?_ga=2.75749783.365316916.1539217252- 1670325792.1539217252 Given the limited observation and the simple modeling, maybe it would have been a good idea to focus on the total city emission. I do not have an access to disaggregated CO2 emissions, but traffic emissions account for 28% of the city total emission, the residential is for 16% and commercial and industrial for 54%. Ignoring the contributions from other sectors does not seem to be a good idea, especially w/o doing any source attribution analysis.

*We agree with the reviewer that it would be nice to focus on the total city emissions. However, the total city emissions are calculated in such a way that we cannot do a direct comparison with our measurements, and as discussed in the updated version given the time of day, time of year, and footprints we are primarily sensitive to traffic.*

*The City of Boulder inventory is Scope 1 and 2 (emissions from inside the city and emissions from electricity usage when the electricity is generated outside of the city). However, our emissions measurements are Scope 1 (emissions at the location of the source). We have done our best to account for power generation that falls within our measurement area. We discuss this is the updated 3.3.3:*

[revised manuscript text omitted]

3. Background problem?

This study used CO2 data from the reference path as a background. I understand that the air must be clean for the reference data, but my concern was the authors were comparing two different airmasses to calculate the CO2 enhancement. The only supporting information of background CO2 vs. city CO2 was the wind direction from a few observation points.

*We have calculated one-hour back trajectories for each hour of the case study days using STILT. These back trajectories indicate that the reference path has a much smaller footprint (as expected) but that the general airmass location is identical for the reference and city paths. Representative back trajectories and footprint calculations are now shown in Figure 7 reproduced earlier. Further, for the measurement conditions, it takes on average 18 minutes for air entering the reference path to cross the over-city path. Since the transit time is so short (i.e., it does not take multiple hours for the air to move from the reference path to the over-city path), it is unlikely that the source location of the airmass is different for the reference and over-city paths.*

*Please see the updated Figure 7 and related text as described in the response to Reviewer 1 above.*

4. CO2-eq.

I do not understand why the authors did not use emissions in the CO2 unit, rather than in the unit of CO2-eq. In collaboration with the city council, I would imagine it is not too difficult to obtain emission estimates solely for CO2.

*We have updated the units to MT CO₂/year as suggested.*

5. Bottom-up vs. Top-down?

The discrepancy between bottom-up vs. top-down estimates are often large, as seen in previous studies. In many cases, the uncertainties associated with the inventory are assumed to be small. In this study, the authors have added a lot of potential errors when mapping the traffic emissions in space (approximated the spatial patterns) and time (scaled up to annual emission). Given that, less convincing than other studies if the authors did discussion just with citing papers.

*As noted earlier, we have updated Section 3.3.4 to discuss the scaling more explicitly and we have included uncertainty associated with this scaling. We point out in Section 4.1 and the Conclusion that a better inventory for the city would be highly beneficial. Please see the revised Sections 3.3.4 and 4.1 above in response to Reviewer 2 Comment 1.*

Other comments:

P1, L31: "top-down measurement" sounds odd to me. How about "top-down approach using atmospheric measurements"?

*We have made the suggested change at line 31.*

P2, L58: A SoCAB CO2 top-down study has appeared on ACPD. Check out Hedelius et al. (2018) ACPD.

*We have added a reference to this work in the new text at lines 54-56:* Data from the Orbiting Carbon Observatory satellite (OCO-2) was recently combined with TCCON data to estimate CO₂ emissions from the LA basin (Hedelius et al., 2018).

P4, L162-: This traffic emission modeling is based on huge assumptions. The errors and uncertainties associated with this modeling needs to be quantified at least to show if the emission estimation approach in this study has a good accuracy to show the utility of the CO2 data. Also, the authors might want to check the consistency/inconsistency between the traffic data the authors used and the new 2016 inventory used.

*We have checked the consistency with the traffic data and the 2016 inventory, please see the response to "Traffic Emissions" above. We have done our best to quantify the uncertainty on our measurements and potential uncertainty due to non-traffic emissions within our measurement area as discussed above. The city does not provide any error estimates on their traffic data but we have added uncertainties as discussed above.*

P6, L238: No correlations is good. . . but these are not shown anywhere.

*We have updated Figure 7 and the associated text in response to a comment by Reviewer 1, please see comments above.*

P8, L321: 5% sounds small, but it is comparable to the potential emission changes we want to detect. I don't think it is insignificant.

*We meant that the 5% value was much smaller than our overall uncertainty. However, this statement has been removed in the revised version of the paper.*

P8, L327-: I think these are another set of big assumptions. These assumptions needed to be tested. The authors cited Gurney et al. (2017), but that is a case for Indianapolis. The authors could use the same logic

for the large discrepancy between the top-down and bottom-up estimates. But the authors' statements are not supported by any quantitative analyses.

*As discussed above, we have expanded this paragraph to become section 3.3.3 on non-traffic sources of $CO_2$ (please see revised text above). To specifically respond to the discussion of biological emissions, leaf senescence is the programmed death of leaves, which occurs when the leaves change color and the chloroplasts die. After senescence has completed, plants no longer contribute to $CO_2$, as they are no longer photosynthesizing. Further, soil respiration is tightly coupled to photosynthesis and when photosynthesis ends, soil respiration decreases significantly. The National Phenology Network tabulates the dates of phenological events, such as leaf buds breaking, leaves turning colors, and leaves falling off trees. The site nearest to Boulder (site 20305, about 64 km north of Boulder) has a falling leaves date of September 15 for box elder trees and October 6 for Eastern cottonwoods in 2016. By late October 2016, we can expect that all of the leaves have fallen off of the trees in Boulder, and thus leaf senescence has completed. We also note that estimated plant contribution to cities varies widely with some cities (Boston, Indianapolis) finding a small but significant contribution from plants and others (Salt Lake) finding no influence from plants. Thus our results fall within the rather large reported plant influence range from the literature. We have modified he text at lines 433-439 to say:*

Once leaf senescence has completed, neither plants nor soil respiration contribute to $CO_2$ signal (Matyssek et al., 2013). The National Phenology Network (USA National Phenology Network, 2018) data shows that for the site nearest to Boulder (64 km north of Boulder), the leaf fall dates were September 15, 2016 for box elder trees October 6, 2016 for Eastern cottonwoods. Thus by our measurement dates leaf senescence should be fully complete and plants will not contribute to the city $CO_2$ enhancement. We note that a wide range of biogenic contributions to $CO_2$ have been noted in the literature (Gurney et al., 2017; Mitchell et al., 2018; Sargent et al., 2018).

P16, Figure 1: This figure needs to be improved. It does not even clearly show the traffic distributions.

*We have modified the figure as suggested by this reviewer and by the other reviewer. To address the traffic distributions, the traffic marker size now scales with traffic count. We hope the new figure is clearer.*

P19, Figure 4: Given the small changes in CO2 we are discussing, I think the range of Y is too large. We can't see the variability in CO2 data.

*As discussed in the response to a similar comment from the other reviewer, we have altered Figure 4 to improve the visibility of the variability.*

---

## Author Response (AR2)

Dear Editor,

We thank the Reviewer for his or her thoughtful and thorough review of our manuscript. We appreciate the time and effort that he or she put into the review and we think that the comments have improved our paper. We hope that we have sufficiently addressed the reviewer comments. We have reproduced the reviewer comments below in black. Our responses follow in blue and changes to the text are in green.

Sincerely,
Eleanor Waxman on behalf of the coauthors

To preface this, I was not one of the original reviewers of this manuscript. In contrast to the rather negative first round of reviews, I quite like this manuscript. I think Waxman et al. do a nice job of presenting an (albeit uncertain) CO2 emissions estimate for Boulder using a novel instrument (dual frequency comb spectrometer). The authors have been quite diligent in their analyses over the years including direct comparisons with a CRDS from NOAA on a tower along their beam in previous work. I think this work should be published in ACP with minor corrections. It is well-written, the figures are high quality, and the analysis is quite novel. In my opinion, the only real short-comings are the lack of a high-quality bottom-up inventory and the fairly simplistic atmospheric modeling. I only have one substantial comment below (and a few editorial comments):

* Comparison of the Gaussian plume with Footprints:
It seems that the Gaussian plume analysis was carried over from the original manuscript and the footprint analysis was conducted in response to the first round of reviewers. However the Gaussian plume setup is fairly simplistic in that it assumes a set of point sources and uniform winds. The authors now have footprints for their time period. It would nice to see a comparison between the concentrations simulated with the Gaussian plume and the footprints. That is to say, could the authors 1) compute the concentrations with their Gaussian plume and 2) run those same emission point sources through their footprints to get the concentrations? Large differences would imply that their Gaussian plume calculation is missing something, if the results are similar then it would give me more confidence in their emissions estimates.

We have done what the reviewer suggests. We ran STILT with the meteorological input from 1) HRRR, 2) NAM, 3) NARR, and 4) measured data. The measured data was from local weather stations (see Figure 1 of the main text for locations). Two modeled meteorological inputs disagreed with the measured winds. HRRR had wind directions routinely inconsistent with the measured wind directions (e.g. winds coming out the north moving to out of the east moving to out of the south during time periods when the measurements showed a consistent westerly wind). NARR winds were consistently higher than observed. Therefore, we do not consider results with either HRRR or NARR reliable. NAM generally agreed with the wind speed and direction but is suboptimal because it is on a 12 km grid with 3-hour averaging. Therefore it is on both a timescale and length-scale that is not well matched with our experiment. For these reasons, we ultimately ran STILT using our measured wind field. To generate the input meteorological file, we used the HYSPLIT user interface to generate a vertical velocity variance by designation of stability class (consistent with the classes used for the Gaussian plume modeling).

We ran STILT varying a number of parameters with the measurement wind fields. The parameter that dominated the emissions calculation from STILT was the vertical mixing (KBLT). When set to 1 (the settings used by Karion et al. 2019, Atmos. Chem. Phys. "Intercomparison of atmospheric trace gas dispersion models: Barnett Shale case study"), we got an emissions value of approximately 560 MT/hour and when set to 4 (the setting that uses the velocity variances from the input meteorological data), we got an emissions value of approximately 55 MT/hour. These emissions values bracket both the city inventory emissions and our value by a factor of three in each direction. We also ran STILT with the NAM input data. In that case, we used the STILT settings used by Karion et al. 2019, although varying these parameters had little effect. This resulted in emission value of approximately 770 MT/hour, but as noted above we do not think the coarse time and spatial resolution of the NAM input fields is sufficient. For all runs, we used the hyper near-field setting (hnf_plume = true).

We attribute the observed factor of almost 10 in range of results from STILT to the sensitivity of the footprint to vertical dispersion, which is important especially at these kilometer-scale ranges. The advantage of the Gaussian plume model is that it is clear what dispersion values are used, though as stated in the paper we have not included an additional uncertainty for the vertical dispersion.

We have added to the paper at line 378:

We further ran plume calculations in STILT-R using both wind fields derived from the local meteorological stations shown in Figure 1 and using the North American Mesoscale Forecast System (NAM, https://www.ncdc.noaa.gov/data-access/model-data/model-datasets/north-american-mesoscale-forecast-system-nam). The High Resolution Rapid Refresh (HRRR, https://rapidrefresh.noaa.gov/hrrr/) and North American Regional Reanalysis (NARR, https://www.ncdc.noaa.gov/data-access/model-data/model-datasets/north-american-regional-reanalysis-narr) wind projections did not match the measured winds at the meteorological stations. These calculations produced emissions values ranging between 55 MT/hour and 770 MT/hour, depending on the wind fields and vertical dispersion parameterization used. This brackets our emissions calculations by approximately a factor of three in each direction and shows how sensitive these kilometer-scale measurements are to vertical dispersion.

We have also added an acknowledgement to Anna Karion for assistance with the STILT modeling.

Line 46: Previous OSSE work has actually shown that the low-cost sensors are less sensitive to systematic biases at individual sites (Turner et al., ACP, 2016) because 1) you have an abundance of sites and 2) because you're putting less weight on any given site since the error is larger.

We thank the reviewer for pointing this out. We have modified the statement at line 46 to read:

The BEACO$_2$N network (Shusterman et al., 2016), on the other hand, has a much lower cost per sensor. It requires calibration for quantitative results, but the high density of the point sensors can provide lower sensitivity to systematics (Turner et al., 2016).

Lines 74-80: It would be good to list the specific papers that show these because the measurements are novel to most atmospheric chemists.

We agree with the reviewer and have cited several works on dual comb spectroscopy in this section.

Line 86: Which figure? The diurnal cycle is more consistent with a morning-build up from traffic, than a mid-day decline due to the rising PBL. I'd say the diurnal cycle is more consistent with meteorology than anthropogenic sources.

We agree and have changed the text to read:

The dry mole fraction of $CO_2$ shows a diurnal cycle consistent with a morning build-up from traffic followed by a mid-day decline due to the rising boundary layer.

Lines 246-254: This discussion of CH4 seemed a bit out of place since the rest of the paper is on CO2, not sure if it adds that much or distracts from the message.

We agree with the reviewer that this section is a little out of place. However, we would prefer to leave this paragraph as we do measure $CH_4$ and show our measurements and it would also seem strange to not discuss those measurements at all.

Line 277: Why use NARR? Isn't HRRR available? That would be much higher resolution and is easy to use with the new STILT

We only use the NARR data to obtain the boundary layer height which has minimal impact on the modeling as the spatial scales are small enough that the plume does not mix up to the boundary layer height (based on the Gaussian plume expansion term $\sigma_z$). The wind fields come from the NCAR meteorological stations as described above.

Eq 4: Writing out the steps in a supplement would be useful. I had to look at this equation a few times before it made sense.

We agree and have added an appendix describing how we got from Equation 1 to Equation 4.

Appendix A: Modification of the Gaussian plume equation

Equation 1 is the standard Gaussian plume equation as discussed in Section 3.3.2 (Seinfeld and Pandis, 2006). It is reproduced here,

$$c(x,y,z,t) = \frac{q}{2\pi\sigma_y\sigma_z u}\exp\left(\frac{-(y-y_0)^2}{2\sigma_y^2}\right)\left[\exp\left(\frac{-(z-H)^2}{2\sigma_z^2}\right) + \exp\left(\frac{-(z+H)^2}{2\sigma_z^2}\right)\right]$$

where the standard variables are as defined in Section 3.3.2.

*Path-integrated substitutions*

The DCS returns the average concentration along a line path. We denote distance along this path by the variable $s$, where $s$ runs from 0 to $L$. This path is assumed to lie in the x-y plane at an angle $\theta$ with respect to the x-axis (which is assumed to be the wind direction in the standard Gaussian plume equation). With these definitions, the contribution to the DCS signal from the plume is,

$$(c - c_0) = \frac{1}{L} \int_0^L c(s \cos\theta, \ s \sin\theta, z, t) ds$$

or:

$$(c - c_0) = \frac{1}{L} \frac{q}{2\pi\sigma_y\sigma_z u} \int_0^L \exp\left(\frac{-(s \sin\theta - y_0)^2}{2\sigma_y^2}\right)\left[\exp\left(\frac{-(z-H)^2}{2\sigma_z^2}\right) + \exp\left(\frac{-(z+H)^2}{2\sigma_z^2}\right)\right] ds$$

*Accounting for multiple point sources*

Rather than a single source at ($x_0$, $y_0$), we have multiple sources at locations ($x_j$, $y_j$), each with a source strength $f_j q$, where $f_j$ is the fractional source strength out of the total value $q$. We now sum over all sources to find the total enhancement. We also change the units of $q$ from kg/s to MT/year and thus change the emissions variable to $Q$ to indicate the unit change. This gives,

$$(c - c_0) = \frac{Q}{L} \sum_{(x_j, y_j)} \int_0^L \frac{f_j}{2\pi\sigma_y\sigma_z u} \exp\left(\frac{-(s \sin\theta - y_j)^2}{2\sigma_y^2}\right)\left[\exp\left(\frac{-(z-H)^2}{2\sigma_z^2}\right) + \exp\left(\frac{-(z+H)^2}{2\sigma_z^2}\right)\right] ds$$

*Height substitutions*

We assume that the point source emissions locations are 1 meter above ground ($z = 1$) and city topographic data indicates that our beam path is approximately 15 meters above ground ($H = 15$). These substitutions finally lead to Eq. (4) in the main text.

Line 380: I think this should be "These non-traffic sources", not "source"

We agree and have fixed this.

[revised manuscript text omitted]